# RaVL: Discovering and Mitigating Spurious Correlations in Fine-Tuned Vision-Language Models

**Maya Varma**
Stanford University
mayavarma@cs.stanford.edu

**Jean-Benoit Delbrouck**
Stanford University; Hugging Face
jbdel@stanford.edu

**Zhihong Chen**
Stanford University
zhihongc@stanford.edu

**Akshay Chaudhari**[†]
Stanford University
akshaysc@stanford.edu

**Curtis Langlotz**[†]
Stanford University
langlotz@stanford.edu

## Abstract

Fine-tuned vision-language models (VLMs) often capture spurious correlations between image features and textual attributes, resulting in degraded zero-shot performance at test time. Existing approaches for addressing spurious correlations (i) primarily operate at the global image-level rather than intervening directly on fine-grained image features and (ii) are predominantly designed for unimodal settings. In this work, we present RaVL, which takes a fine-grained perspective on VLM robustness by discovering and mitigating spurious correlations using local image features rather than operating at the global image level. Given a fine-tuned VLM, RaVL first **discovers** spurious correlations by leveraging a region-level clustering approach to identify precise image features contributing to zero-shot classification errors. Then, RaVL **mitigates** the identified spurious correlation with a novel region-aware loss function that enables the VLM to focus on relevant regions and ignore spurious relationships during fine-tuning. We evaluate RaVL on 654 VLMs with various model architectures, data domains, and learned spurious correlations. Our results show that RaVL accurately discovers (191% improvement over the closest baseline) and mitigates (8.2% improvement on worst-group image classification accuracy) spurious correlations. Qualitative evaluations on general-domain and medical-domain VLMs confirm our findings.[1]

## 1 Introduction

Contrastive vision-language models (VLMs) (e.g., CLIP [36] and ALIGN [24]) are a powerful class of models that jointly learn relationships between images and text. VLMs are generally pretrained on web-scale datasets with millions of image-text pairs and have been shown to exhibit impressive capabilities on a wide range of downstream tasks. In particular, VLMs have the ability to perform tasks in a zero-shot manner without utilizing explicit task-specific training data; this is accomplished by modeling downstream tasks (e.g., image classification, text-to-image retrieval) as image-text matching tasks [36].

However, pretrained VLMs can exhibit poor zero-shot performance when compared to state-of-the-art task-specific models, particularly on challenging or out-of-domain downstream tasks [36, 7, 17, 19]. As a result, pretrained VLMs are often fine-tuned on domain-specific vision-language datasets in order

---

[†]Equal senior authorship.
[1]Code: https://github.com/Stanford-AIMI/RaVL

38th Conference on Neural Information Processing Systems (NeurIPS 2024).

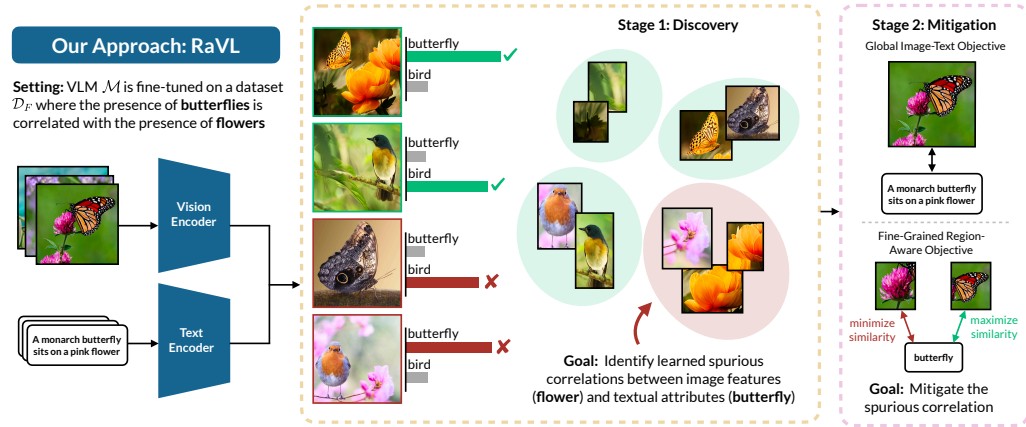

Figure 1: *Region-aware Vision-Language learning (RAVL).* RAVL takes a fine-grained perspective on VLM robustness by discovering and mitigating spurious correlations using local image features.

to improve zero-shot performance on tasks of interest. For instance, recent works have fine-tuned the CLIP VLM [36] on vision-language datasets consisting of (i) chest X-rays and paired physician reports [45], (ii) pathology data and paired text [17, 19], and (iii) product images and paired captions from online fashion retailers [7].

Domain-specific vision-language datasets used to fine-tune VLMs may be small in size, preventing VLMs from gaining the robustness benefits that come with training on diverse, web-scale data [6, 14]. As a result, fine-tuned VLMs may capture spurious correlations between image features and textual attributes [56]. For instance, consider a VLM fine-tuned on an animal image-text dataset where the presence of butterflies is closely correlated with the presence of flowers (Figure 1). Consequently, the VLM may learn to incorrectly associate the image features corresponding to *flower* with the textual attribute *butterfly*. At test time, the VLM is likely to exhibit degraded zero-shot classification performance on (i) images of butterflies without flowers and (ii) images of other animals with flowers.

Improving robustness of fine-tuned VLMs to spurious correlations is challenging for the following two reasons. First, existing automated approaches primarily discover and mitigate spurious correlations at the global image level rather than intervening directly on fine-grained image features. Such approaches discover spurious correlations by identifying coherent groups of misclassified images in an automated fashion [13, 43, 22, 42]; then, the identified spurious correlation can be mitigated during training using data augmentation or robust optimization [43, 39, 22, 56]. However, recent works have suggested that such global image-level strategies (i) discover spurious correlations that align poorly with human-interpretable attributes [25] and (ii) may not effectively enable models to ignore spurious correlations during training [15, 18]. Second, existing approaches for discovering and mitigating spurious correlations are predominantly designed to improve robustness of unimodal image classification models [39, 43] or pretrained VLMs [60, 49]. These settings differ substantially from the fine-tuned VLM setting, which presents several unique challenges such as the absence of class and subgroup labels in the training set and the inclusion of free-form text.

In this work, we address these challenges by introducing **R**egion-**a**ware **V**ision-**L**anguage learning (RAVL), an approach for improving the robustness of fine-tuned VLMs to spurious correlations. RAVL takes a *fine-grained* perspective on VLM robustness by discovering and mitigating spurious correlations using local image features, rather than operating at the global image level. Our contributions are:

- First, given a fine-tuned VLM, RAVL **discovers** learned spurious correlations between image features and textual attributes. Using a labeled classification dataset, we decompose images into candidate regions, utilize the VLM embedding space to group visually-similar regions into feature clusters, and quantitatively evaluate the effects of each feature on zero-shot classification errors.
- Second, given a ranked list of image features that the VLM has learned to spuriously correlate with one or more textual attributes, RAVL **mitigates** the identified spurious correlations. Our key insight is that region-level information can be leveraged during VLM fine-tuning in order to improve model robustness. To this end, we introduce a novel region-aware loss function

that encourages the VLM to focus on relevant regions and ignore spurious relationships during fine-tuning.

In order to evaluate RAVL, we introduce a large-scale evaluation framework for controlled, fine-grained evaluations of VLM robustness on synthetic and real-world data. Our framework consists of 654 fine-tuned VLMs paired with annotations for the ground-truth spurious correlations learned by each VLM. Across these evaluation settings, (i) RAVL accurately discovers spurious correlations, achieving a 191% improvement over the closest baseline, and (ii) RAVL effectively mitigates spurious correlations, achieving up to an 8.2% improvement on worst-group image classification accuracy. Qualitative evaluations on general-domain and medical-domain VLMs confirm the utility of RAVL.

This paper is organized as follows. In Section 2, we introduce our problem setting. Then, in Section 3, we present Stage 1 of RAVL, including our proposed methodology for discovering spurious correlations, our large-scale evaluation framework, and experimental results. In Section 4, we introduce Stage 2 of RAVL, including our proposed methodology for mitigating spurious correlations as well as experimental results. Finally, we conclude in Section 5.

**Related Work**: Our work builds on several recent research directions for discovering and mitigating spurious correlations. We provide an analysis of related works in Appendix Section A.

## 2   Preliminaries

In this section, we formally describe our problem setting. Datasets used for fine-tuning VLMs can be expressed as $\mathcal{D}_F = \{(I_i, T_i)\}_{i=1}^m$, where $I_i$ represents image inputs and $T_i$ represents paired free-form text. We do not assume access to any class or subgroup labels.

The performance of fine-tuned VLMs can be characterized with zero-shot classification tasks. In line with prior work [13, 22, 56], we assume that the zero-shot classification dataset includes a validation split $\mathcal{D}_V = \{(I_i, y_i)\}_{i=1}^n$ with images $I_i$ and known ground-truth class labels $y_i \in \mathcal{Y}$, where $\mathcal{Y}$ denotes the set of all possible class labels. At evaluation time, classification performance is computed by encoding class labels in $\mathcal{Y}$ as text and matching images to the closest class label using embedding similarity. We do not assume access to any subgroup labels.

Fine-tuned VLMs may learn spurious correlations between image features and textual attributes. Let $\mathbf{e}_a$ represent the image features corresponding to a visual concept $a$ (e.g., flowers in Figure 1) and $y \in \mathcal{Y}$ represent a class label (e.g., "butterfly" in Figure 1) such that $\mathbf{e}_a$ and $y$ share no causal relationship. Then, a fine-tuned VLM that has learned a spurious correlation will be unable to disentangle $\mathbf{e}_a$ and $y$ at evaluation time. This will manifest in low zero-shot classification performance on the following two subgroups of data: (i) images from class label $y$ without the feature $\mathbf{e}_a$ and (ii) images from other class labels $\mathcal{Y} \setminus \{y\}$ with the feature $\mathbf{e}_a$.

However, since neither the fine-tuning dataset $\mathcal{D}_F$ nor the evaluation dataset $\mathcal{D}_V$ include subgroup labels corresponding to visual concepts $a$, discovering and mitigating such spurious correlations poses a challenge. For instance, in Figure 1, there are no annotations for flowers in datasets $\mathcal{D}_F$ and $\mathcal{D}_\mathcal{V}$, making it challenging to identify and address the learned spurious correlation between image features corresponding to flowers and the textual attribute corresponding to "butterfly".

In the following sections, we will discuss our automated approach RAVL, which aims to address this challenge by employing fine-grained region-level information to discover (Section 3) and mitigate (Section 4) spurious correlations in fine-tuned vision-language models.

## 3   Discovering Spurious Correlations in Fine-Tuned Vision-Language Models

In this section, we present the first stage of RAVL, which aims to discover learned spurious correlations in VLMs. In Section 3.1, we discuss our region-aware approach for discovering fine-grained spurious correlations. Then, in order to quantitatively evaluate the efficacy of spurious feature discovery methods, we introduce a large-scale evaluation framework in Section 3.2. Finally, in Section 3.3, we use our evaluation framework to demonstrate that RAVL outperforms prior approaches in discovering fine-grained spurious correlations between image features and textual attributes.

## 3.1 Our Approach: Discovering Spurious Correlations

The first stage of RaVL aims to identify spurious correlations between image features and textual attributes learned by a fine-tuned VLM $\mathcal{M}$. In contrast to prior works that have incorporated humans in the loop in order to identify spurious correlations [56, 30], RaVL is a fully automated approach. Additionally, whereas previous automated methods for discovering spurious correlations focus predominantly on identifying groups of *images* with high error rates [22, 13], our approach identifies specific *image features* that model $\mathcal{M}$ has learned to spuriously correlate with a textual attribute. Our goal is to discover precise spurious correlations that can be easily interpreted by humans.

As discussed in Section 2, a model $\mathcal{M}$ that has learned a spurious correlation between an image feature $\mathbf{e}_a$ and a textual attribute $y$ will demonstrate low zero-shot performance on (i) images in $\mathcal{D}_V$ with label $y$ without the feature $\mathbf{e}_a$ and (ii) images in $\mathcal{D}_V$ with other labels $\mathcal{Y} \setminus \{y\}$ with the feature $\mathbf{e}_a$. The key challenge lies in identifying such relationships when no annotations are provided for visual concepts $a$. RaVL addresses this challenge by (1) obtaining candidate image features in $\mathcal{D}_V$, (2) identifying the candidate image features that, when present in an image, directly contribute to classification errors, and (3) ranking the identified image features by degree of learned spurious correlations.

**Obtaining candidate image features.** RaVL first utilizes the zero-shot classification dataset $\mathcal{D}_V$ to identify candidate image features. To this end, we use the fine-tuned VLM $\mathcal{M}$ to extract an image embedding for each image $I_i$ in $\mathcal{D}_V$ and a text embedding for each class $y \in \mathcal{Y}$. Zero-shot classification is performed using the computed embeddings; this results in a softmax-normalized image score distribution vector $\mathbf{s}_{I_i} \in \mathbb{R}^{|\mathcal{Y}|}$, where $|\mathcal{Y}|$ represents the number of classes. Then, we decompose each image $I_i$ in $\mathcal{D}_V$ into a set of candidate *regions* $\mathcal{R}_i$. There are a variety of ways in which an image can be decomposed into regions, such as dividing images into equal-sized segments (e.g., quadrants) or using region proposal networks (RPNs) [38]. Ideally, regions should capture key features in the image; however, **we emphasize that RaVL does not require ground-truth region-level annotations**. We then apply RoIAlign [16, 63] to the image encoder of $\mathcal{M}$ to extract embeddings for each region. Zero-shot classification is performed using the computed region embeddings, resulting in a softmax-normalized region score distribution matrix $\mathbf{S}_{R_i} \in \mathbb{R}^{|\mathcal{R}_i| \times |\mathcal{Y}|}$.

Given region-level embeddings for all candidate regions in $\mathcal{D}_V$, we next aim to identify coherent groups of image features that occur consistently throughout the dataset (e.g., features corresponding to "flower" or "butterfly" in Figure 1). To this end, we cluster the computed region-level embeddings using the K-Medoids algorithm with cosine distance. The optimal number of clusters is selected in an automated fashion using Silhouette distance. The resulting clusters (denoted as $\mathcal{C}$) capture key image features in $\mathcal{D}_V$. For feature cluster $c \in \mathcal{C}$, let $\mathbf{e}_c$ denotes the set of features in cluster $c$.

**Identifying candidate image features that directly contribute to classification errors.** We now seek to identify features that, when present in an image, are directly responsible for prediction errors.

Let $\mathcal{R}_c$ represent the set of regions assigned to cluster $c$ and let $\mathcal{I}_c$ represent the set of images associated with the regions in cluster $c$. We identify labels for images in $\mathcal{I}_c$; we designate this label set as $\mathcal{Y}_c$. For each class label $y \in \mathcal{Y}_c$, we identify all images in $\mathcal{I}_c$ with label $y$, and we designate zero-shot classification accuracy on this subset of $n_{in}^y$ images as $p_{in}^y$. Then, we identify all images in $\mathcal{D}_v$ with label $y$ that do not have a region included in cluster $c$, and we designate zero-shot classification accuracy on this subset of $n_{out}^y$ images as $p_{out}^y$.

We now introduce the *cluster influence score*, which evaluates the extent to which features $\mathbf{e}_c$ contribute to mispredicted image classification labels. We restrict our evaluation to only include mispredicted images in $\mathcal{I}_c$ with ground-truth labels $y$ such that $p_{in}^y < p_{out}^y$; we will refer to this subset as $\mathcal{I}_c^{err} \subset \mathcal{I}_c$. For each image $I_i \in \mathcal{I}_c^{err}$, we extract (i) the image score distribution vector $\mathbf{s}_{I_i}$ and (ii) the region score distribution matrix $\mathbf{S}_{R_i}$. We use $\mathbf{s}_{I_i}$ to identify the predicted image class $\hat{y}$, and we then identify the region $r_i^{max}$ in $\mathcal{R}_i$ with the highest score for class $\hat{y}$.

*Definition 1 (Cluster Influence Score).* For cluster $c$ and label $y$, the cluster influence score is the proportion of images $I_i \in \mathcal{I}_c^{err}$ with label $y$ where the identified highest-scoring region $r_i^{max}$ is part of cluster $c$ (i.e., $r_i^{max} \in \mathcal{R}_c$):

$$H_c^y = \frac{1}{|\{I_i \in \mathcal{I}_c^{err} | y_i = y\}|} \sum_{I_i \in \mathcal{I}_c^{err}; y_i = y} \mathbb{1}[r_i^{max} \in \mathcal{R}_c] \tag{1}$$

The final cluster influence score for cluster $c$ is computed as the maximum over all labels $y$ as $H_c = max_{y \in \mathcal{Y}_c} H_c^y$. High values of $H_c$ show that features $\mathbf{e}_c$ are similar to the incorrect label in the vision-language embedding space; this suggests that for a given image with an incorrect prediction, feature $\mathbf{e}_c$ is more likely to contribute to the misprediction than other features in the image. On the other hand, low values of $H_c$ are likely to indicate that feature $\mathbf{e}_c$ represents a core feature associated with the class label or a neutral feature that does not affect predictions.

Given $H_c$ for each feature cluster, we prune all clusters with influence scores below a threshold of $\tau_l$, which we set to 0.25 in all experiments.

**Ranking image features by degree of learned spurious correlation.** For each remaining feature cluster, we next aim to determine the extent to which the presence or absence of features $\mathbf{e}_c$ affects classification performance; we introduce the *cluster performance gap* metric to this end.

*Definition 2 (Cluster Performance Gap).* For cluster $c$ and label $y$, the cluster performance gap is the weighted difference between zero-shot classification accuracy on images with features $\mathbf{e}_c$ and images without features $\mathbf{e}_c$:

$$G_c^y = w_y \times (p_{in}^y - p_{out}^y), \tag{2}$$

where $w_y$ is a simple weighting factor computed as $w_y = 2 \times [\min(n_{in}^y, n_{out}^y)/(n_{in}^y + n_{out}^y)]$.

Since spurious correlations result in consistent errors as opposed to isolated misclassifications, the weighting factor is designed to prioritize stronger spurious correlations that result in a larger number of errors. $G_c^y$ ranges between 0 and 1. The final performance gap metric for cluster $c$ is computed across all labels as $G_c = \sum_{y \in \mathcal{Y}_c} |G_c^y|$. A high value of $G_c$ suggests that the presence or absence of features $\mathbf{e}_c$ contribute to large class-level variations in image classification performance.

Given $G_c$ for each feature cluster, we rank clusters in order from highest to lowest values. The output of this stage is a ranked list of image features that model $\mathcal{M}$ has learned to spuriously correlate with one or more class labels in $\mathcal{Y}$.

### 3.2 Experimental Setup: Designing a Large-Scale Evaluation Framework

We now discuss our approach for evaluating RAVL. Evaluating the accuracy of predicted spurious correlations is challenging because the ground-truth spurious correlations learned by a model $\mathcal{M}$ are typically unknown. Previous works on VLM robustness evaluate discovered spurious correlations with qualitative experiments, human-in-the-loop evaluations, or small-scale datasets [56]. Our aim in this section is to introduce a large-scale experimental setup where the ground-truth spurious correlations learned by VLMs are known and annotated in advance; this can then enable us to determine whether the features discovered by RAVL in Section 3.1 accurately align with the ground-truth. Our evaluation framework is motivated by prior work [26, 13]; however, in contrast to existing approaches, we introduce evaluation settings that are designed (i) for evaluating robustness approaches at the fine-grained region level rather than the global image-level, and (ii) for evaluating VLMs rather than unimodal models.

**Designing Controlled Evaluations:** Our evaluation framework artificially induces spurious correlations in the VLM fine-tuning data; then, given the known pre-defined spurious correlation and a VLM that learned the desired spurious correlation, we can quantitatively evaluate the extent to which RaVL discovers the correlation.

We create a set of *evaluation settings* using data from two domains: (1) synthetic data (MNIST [11] and FashionMNIST [51]) and (2) real-world data (COCO [27]). Each evaluation setting consists of the following components:

1. *Predefined spurious correlation*: We define a spurious image feature and textual attribute pair ($\mathbf{e}^{eval}$, $a^{eval}$). For MNIST and FashionMNIST, $\mathbf{e}^{eval}$ represents a red rectangle; $a^{eval}$ is generated from the set of class labels {zero, one, two, three, four five, six, seven, eight, nine} for MNIST and {t-shirt, trouser, pullover, dress, coat, sandal, shirt, sneaker, bag, ankle boot} for FashionMNIST. For COCO, we sample $\mathbf{e}^{eval}$ and $a^{eval}$ from the list of annotated attributes.
2. *Fine-tuning dataset*: We construct a vision-language fine-tuning dataset $\mathcal{D}_F^{eval} = \{(I_i, T_i)\}_{i=1}^m$ with images $I_i$ and text $T_i$. Dataset $\mathcal{D}_F^{eval}$ is sampled from the training sets of MNIST, Fashion-MNIST, or COCO such that the presence of image feature $\mathbf{e}^{eval}$ is closely correlated with the presence of text attribute $a^{eval}$ as measured by Cramer's V [57].

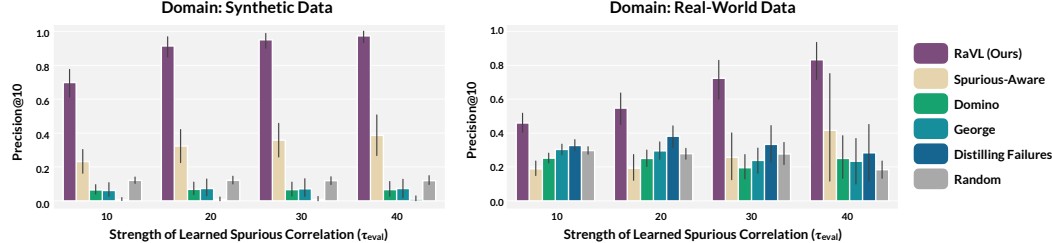

Figure 2: RAVL *accurately identifies spurious correlations.* Using our evaluation settings, we show that RAVL consistently outperforms prior methods in discovering learned spurious correlations between image features and textual attributes. Here, we provide Precision@10 metrics for a CLIP-RN50 model fine-tuned on synthetic data (129 settings) and real-world data (171 settings).

Table 1: *Mean Precision@10 metrics demonstrate the efficacy of* RAVL *in discovering spurious correlations.* On average across 654 evaluation settings, RAVL consistently outperforms baselines.

| Method | Correlation Strength ($\tau_{eval}$) | | | |
|---|---|---|---|---|
| | 10 | 20 | 30 | 40 |
| Num. Eval Settings | 654 | 369 | 234 | 168 |
| Random | 21.2 | 18.2 | 15.5 | 12.5 |
| Distilling Failures | 20.1 | 16.2 | 8.5 | 1.5 |
| George | 19.3 | 15.9 | 10.9 | 7.7 |
| Domino | 17.1 | 15.0 | 11.7 | 9.0 |
| Spurious-Aware Detection | 20.0 | 25.3 | 32.1 | 42.0 |
| RAVL (Ours) | **61.8** | **76.2** | **84.2** | **91.1** |

Table 2: *Ablations show the utility of the cluster performance gap and influence metrics.* We report Precision@10 metrics for a CLIP-RN50 model fine-tuned on real-world data (171 settings).

| Ablation | Correlation Strength ($\tau_{eval}$) | | | |
|---|---|---|---|---|
| | 10 | 20 | 30 | 40 |
| Unweighted $G_c$ Only | 21.2 | 30.0 | 36.2 | 55.0 |
| $G_c$ Only | 40.9 | 51.7 | 63.8 | 66.7 |
| $G_c$ & $H_c$ (RAVL) | **46.0** | **54.8** | **72.4** | **83.3** |

3. *Fine-tuned VLM*: A VLM $\mathcal{M}$ is fine-tuned on $\mathcal{D}_F^{eval}$.

4. *Evaluation dataset*: Model $\mathcal{M}$ is evaluated using a zero-shot classification dataset $\mathcal{D}_V^{eval} = \{(I_i, y_i, \mathcal{R}_i, \mathcal{L}_i)\}_{i=1}^n$ with images $I_i$, class labels $y_i$, region bounding boxes $\mathcal{R}_i$, and region-level labels $\mathcal{L}_i$. In particular, $a^{eval}$ must be included in the class label set, and $\mathbf{e}^{eval}$ must be annotated in the region-level label set. Since $\mathcal{D}_V^{eval}$ is designed to reflect a real-world setting, we assume that a correlation between $a^{eval}$ and $\mathbf{e}^{eval}$ does not exist. Dataset $\mathcal{D}_V^{eval}$ is constructed from the test sets of MNIST, FashionMNIST, or COCO.

Given the four components listed above, we classify an evaluation setting as valid if model $\mathcal{M}$ learned the intended spurious correlation. In order to measure this, we first identify images with label $a^{eval}$ in $\mathcal{D}_V^{eval}$ and compute the performance difference between images with feature $\mathbf{e}^{eval}$ and images without feature $\mathbf{e}^{eval}$; we designate this value as $\epsilon_1$. Then, for labels $y \neq a^{eval}$, we compute the maximum performance difference between images without feature $\mathbf{e}^{eval}$ and images with feature $\mathbf{e}^{eval}$; we designate this value as $\epsilon_2$. Large values of $\epsilon_1$ and $\epsilon_2$ suggest that model $\mathcal{M}$ has learned the desired spurious correlation between image feature $\mathbf{e}^{eval}$ and textual attribute $a^{eval}$, as defined in Section 2. We remove settings where $\epsilon_1$ or $\epsilon_2$ are below some predefined performance threshold $\tau_{eval}$. The performance threshold $\tau_{eval}$ serves as a quantitative indicator of learned correlation strength.

**Implementation Details:** In total, we generate 620 fine-tuning datasets $D_F^{eval}$ (100 synthetic; 520 real-world). We then fine-tune model $\mathcal{M}$ on each dataset with three random seeds, resulting in 1860 candidate evaluation settings. Finally, we filter out settings where model $\mathcal{M}$ does not consistently learn the spurious correlation; to this end, we only retain settings where both $\epsilon_1$ and $\epsilon_2$ exceed $\tau_{eval} = 10$ across all three random seeds. We repeat this procedure across various pretrained VLMs $\mathcal{M}$, resulting in 654 valid experimental settings. Additional implementation details are provided in Appendix B.

### 3.3    Results: RaVL Effectively Discovers Spurious Correlations

**Comparisons to Prior Approaches**: Given an evaluation setting with a predefined spurious correlation $(\mathbf{e}^{eval}, a^{eval})$, a fine-tuned VLM $\mathcal{M}$, and an evaluation dataset $\mathcal{D}_V^{eval}$, our goal is to determine the extent to which RaVL can discover the correlation between $\mathbf{e}^{eval}$ and $a^{eval}$.

To this end, we use the labeled zero-shot classification dataset $\mathcal{D}_V^{eval}$, which includes ground-truth region bounding boxes and associated region labels. We provide the ground-truth bounding boxes as input to RaVL, which returns a single top-ranked cluster of regions likely to include spurious features. We rank regions within the cluster based on similarity to the cluster medoid, and we utilize the provided region-level labels in $\mathcal{D}_V^{eval}$ to evaluate the proportion of top-$K$ regions that contain the desired spurious feature $\mathbf{e}^{eval}$. In line with prior work [13], we report performance with Precision@K metrics. We note that given an identified spurious feature $\mathbf{e}^{eval}$, the correlated textual attribute $a^{eval}$ can be detected by identifying the class label in $\mathcal{D}_V^{eval}$ where the absence of feature $\mathbf{e}^{eval}$ leads to degraded performance.

There are few existing approaches for performing automated detection of fine-grained spurious features learned by VLMs. Here, we compare RaVL with four previously-developed methods: Distilling Failures [22], George [43], Domino [13], and Spurious-Aware Detection [56]. Distilling Failures, George, and Domino are state-of-the-art approaches for automatic identification of model failures resulting from spurious correlations; although these methods operate at the global image level and are designed for unimodal settings, we adapt these approaches for our setting by utilizing regions and zero-shot classification scores as input. Spurious-Aware Detection operates at the fine-grained region level by computing class-based performance gaps resulting from the presence or absence of particular features. To enable a fair comparison with RaVL, we provide the same set of regions and associated embeddings as input to all baselines. We also compare RaVL with a random baseline, where the ranked list of regions is shuffled randomly.

Table 1 summarizes mean Precision@10 metrics across all 654 evaluation settings. Results demonstrate that RaVL consistently outperforms prior approaches in discovering spurious correlations between image features and textual attributes, contributing to a 191% improvement over the closest baseline. In Table 1, we evaluate the effects of learned spurious correlation strength by varying the error threshold $\tau_{eval}$ from 10 to 40 and reporting performance for the subset of valid evaluation settings. Results show that RaVL is particularly effective when VLM $\mathcal{M}$ learns a strong spurious correlation; as learned correlation strength increases, performance of RaVL increases by 47% whereas most baselines degrade in performance. We also observe that Domino, George, and Distilling Failures often achieve performance near or below the random baseline across our evaluation settings; this suggests that methods designed for detecting errors resulting from spurious correlations at the global image-level cannot be easily adapted for fine-grained region-level discovery. Figure 2 demonstrates that our findings hold for both synthetic and real-world data.

**Ablations**: Our ablation study evaluates the role of the cluster influence score $H_c$ and the cluster performance gap metric $G_c$ (Section 3.1) in enabling accurate discovery of spurious correlations between image features and textual attributes. We compare the following three metrics for ranking clusters: (1) an unweighted cluster performance gap metric where $w_y$ is set to 1, (2) the cluster performance gap with $w_y$ computed as in Section 3.1, and (3) a combination of the cluster performance gap and cluster influence metric as used in RaVL. As shown in Table 2, the metrics utilized by RaVL consistently demonstrate the best performance across various learned correlation strengths ($\tau_{eval}$). Our results suggest the utility of both the performance gap metric and the influence score in identifying fine-grained spurious correlations.

**Evaluations in the Wild**: In addition to our controlled evaluations, we evaluate the ability of RaVL to surface spurious correlations learned by 12 off-the-shelf VLMs [12, 36, 20]; this presents a realistic and uncontrolled evaluation setting. We consider two zero-shot classification tasks $\mathcal{D}_V$: (1) a 397-class scene classification task on SUN397 [52] and (2) binary classification of cardiomegaly in chest X-rays from ObjectCXR [23]. We use the cluster performance gap metric $G_c$, introduced in Section 3.1, to quantify the degree of the learned spurious correlation.

Our results demonstrate that all evaluated models, which span a range of architecture, training data, and parameter counts, show evidence of having learned spurious correlations; this is demonstrated by nonzero values of the cluster performance gap metric $G_c$. On average across the evaluated models, the top-ranked spurious feature cluster discovered by RaVL on SUN397 achieves a cluster performance

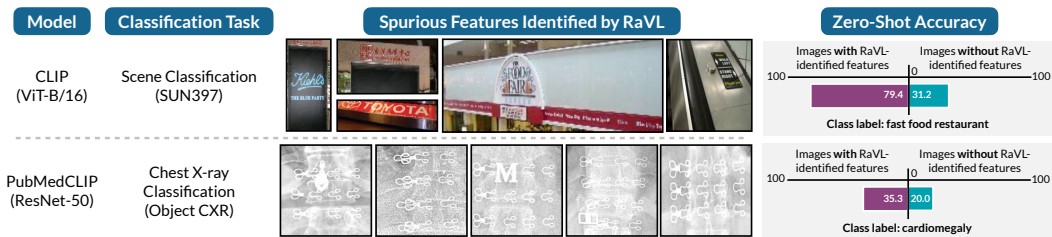

Figure 3: RAVL *surfaces spurious correlations in off-the-shelf VLMs.* RAVL identifies a spurious correlation learned by CLIP ViT-B/16 between the presence of text-based retail signage and the class label `fast food restaurant` in a scene classification task. RAVL also surfaces a spurious correlation learned by PubMedCLIP ResNet-50 between metal clips (found in clothing) and the class label `cardiomegaly` (a heart condition) on a chest X-ray classification task.

Table 3: RAVL *effectively mitigates spurious correlations.* Here, we report mean Image Overall, Image Worst-Group (Img. WG), Region Overall, and Region Worst-Group (Reg. WG) metrics across our real-world evaluation settings. Since performance of mitigation methods is dependent on the results of Stage 1, we report metrics across settings where Stage 1 Precision@10 $> 0.6$ and Stage 1 Precision@10 $> 0.8$.

| Method | Stage 1 Discovery Precision@10 $> 0.6$ | | | | Stage 1 Discovery Precision@10 $> 0.8$ | | | |
|---|---|---|---|---|---|---|---|---|
| | Img. Overall | Img. WG | Reg. Overall | Reg. WG | Img. Overall | Img. WG | Reg. Overall | Reg. WG |
| Standard FT | 64.0 | 31.4 | 72.0 | 46.9 | 64.6 | 31.0 | 72.9 | 47.4 |
| Upsampled FT | 66.6 | 37.8 | 74.3 | 52.2 | 66.7 | 37.7 | 74.7 | 52.8 |
| VL-ERM | 68.8 | 32.2 | 75.6 | 50.3 | 68.7 | 30.9 | 75.9 | 50.6 |
| VL-GDRO | 69.1 | 33.7 | 75.6 | 50.4 | 68.8 | 31.1 | 76.0 | 51.0 |
| Spurious-Aware | **69.8** | 33.6 | 76.5 | 50.6 | 69.2 | 30.7 | 76.8 | 50.5 |
| RAVL (Ours) | **69.8** | **39.1** | **78.9** | **57.8** | **70.2** | **40.8** | **79.5** | **58.5** |

gaps ($G_c$) of $9.9_{\pm 3.2}$ (minimum = 5.1, maximum = 14.0). On ObjectCXR, the mean value of $G_c$ is $0.08_{\pm 0.04}$ (minimum = 0.04, maximum = 0.12).[2] Our results support findings from previous work suggesting that *all* models may learn spurious correlations [30].

In Figure 3, we provide qualitative examples of discovered spurious features for the CLIP ViT-B/16 model evaluated on SUN397 and the PubmedCLIP ResNet-50 model evaluated on ObjectCXR. For the CLIP ViT-B/16 model, RAVL surfaces a feature cluster consisting of text-based retail signage. We observe significant performance gaps between images containing the RAVL-identified feature and images that do not contain the feature. For instance, we note a 48.2 point difference in zero-shot classification accuracy for the class label `fast food restaurant`, suggesting that a CLIP ViT-B/16 model can better classify a scene of a fast food restaurant when a text-based retail sign is present. For the PubmedCLIP ResNet-50 model, RAVL discovers that the presence of metal clips (found in the patient's clothing) is spuriously correlated with cardiomegaly. We observe that the presence of clips improves zero-shot classification accuracy for the class label `cardiomegaly` by 15.3 points.

Our evaluations show that RAVL can surface fine-grained spurious correlations in realistic settings. Additional implementation details and qualitative examples are provided in Appendix D.

# 4 Mitigating Spurious Correlations in Fine-Tuned Vision-Language Models

In this section, we present the second stage of RAVL, which aims to mitigate learned spurious correlations in VLMs. In Section 4.1, we discuss our methodology for mitigating fine-grained spurious correlations with a novel region-aware loss function. In Section 4.2, we use the evaluation framework previously introduced in Section 3.2 to demonstrate that RAVL substantially outperforms prior approaches in mitigating spurious correlations between image features and textual attributes.

---

[2]We note that since the formula for $G_c$ involves a summation over class labels, raw values of $G_c$ for our 2-class chest X-ray classification task are lower than those for our 397-class scene classification task.

## 4.1 Our Approach: Mitigating Spurious Correlations

As described in Section 3, Stage 1 of RAVL discovers image features that VLM $\mathcal{M}$ has learned to spuriously correlate with textual attributes. We next aim to mitigate the spurious correlation. Motivated by prior work on fine-grained VLMs [58, 46], our key insight is that utilizing region-level information during VLM training can enable models to focus on relevant image-text relationships and ignore spurious correlations.

Since dataset $\mathcal{D}_F$ exclusively consists of images and text, ground-truth subgroup and class labels are not available. As a result, we first assign plausible (i) region-level subgroup labels and (ii) image-level class labels to the vision-language fine-tuning dataset $\mathcal{D}_F$. To assign subgroup labels, we decompose each image $I_i$ in dataset $\mathcal{D}_F$ into a set of candidate regions $\mathcal{R}_i$. We then fit the trained K-Medoids clustering model from Section 3.1 on $\mathcal{R}_i$ and identify all spurious regions associated with the top ranked cluster. We represent the identified spurious regions as $\mathcal{R}_i^s$ and remaining non-spurious regions as $\mathcal{R}_i^r$ such that $\mathcal{R}_i^s \cup \mathcal{R}_i^r = \mathcal{R}_i$. In order to assign plausible class labels, we parse the paired text $T_i$ associated with each image to identify samples that reference the class labels included in the zero-shot classification label set $\mathcal{Y}$; we refer to the assigned class label for image $I_i$ as $\hat{y}_i$.

We now introduce a novel region-aware contrastive loss function for training VLM $\mathcal{M}_{new}$. For batch $\mathcal{B}$, we define $\mathcal{R}_\mathcal{B}^s$ as the set of all spurious regions in the batch: $\mathcal{R}_\mathcal{B}^s = \bigcup_{I_i \in \mathcal{B}} \mathcal{R}_i^s$. For image $I_i \in \mathcal{B}$, the first loss component $L_R^i$ encourages high embedding similarity between non-spurious regions $\mathcal{R}_i^r$ and assigned class label $\hat{y}_i$ when compared to other class labels.

$$L_R^i = -\log \frac{\sigma_m(\mathcal{R}_i^r, \hat{y}_i)}{\sum_{\hat{y}_j \in \mathcal{B}} \sigma_m(\mathcal{R}_i^r, \hat{y}_j) + P(\mathcal{R}_\mathcal{B}^s)} \tag{3}$$

Here, for region embedding function $f$ and text embedding function $g$, $\sigma_m(A, b) = \exp(\max_{a \in A}(\langle f(a), g(b) \rangle / \tau))$ with temperature $\tau$. The term $P(\mathcal{R}_\mathcal{B}^s)$ is a penalty that enforces embedding-level dissimilarity between spurious regions and correlated class labels.

The second loss component $L_A^i$ encourages high embedding similarity between non-spurious regions $\mathcal{R}_i^r$ and assigned class label $\hat{y}_i$ when compared to other regions. We define $\sigma(a, b) = \exp(\langle f(a), g(b) \rangle / \tau)$ with temperature $\tau$.

$$L_A^i = -\log \frac{\sigma_m(\mathcal{R}_i^r, \hat{y}_i)}{\sigma_m(\mathcal{R}_i^r, \hat{y}_i) + \sum_{j=1, \hat{y}_j \neq \hat{y}_i}^{|\mathcal{B}|} \sigma_m(\mathcal{R}_j^r, \hat{y}_i) + \sum_{r_j \in \mathcal{R}_\mathcal{B}^s} \sigma(r_j, \hat{y}_i)}. \tag{4}$$

The final loss is expressed as $L = \lambda L_{CL} + (1 - \lambda) \sum_{i=1}^{|\mathcal{B}|} (L_R^i + L_A^i)$. Here, $\lambda$ is a hyperparameter and $L_{CL}$ takes the form of the original loss function used for training $\mathcal{M}$; in our experiments, $L_{CL}$ is the CLIP objective [36]. Extended formulations of our loss function are provided in Appendix C.

## 4.2 Results: RaVL Effectively Mitigates Spurious Correlations

**Comparisons to Prior Approaches**: We use the evaluation framework previously introduced in Section 3.2 to compare RAVL with prior approaches. There are few existing approaches for mitigating spurious correlations in the setting of fine-tuned VLMs. Here, we compare RAVL with standard VLM fine-tuning, upsampled VLM fine-tuning, ERM, GDRO [39], and Spurious-Aware Mitigation [56]. Since ERM and GDRO are traditionally used in unimodal classification settings, we adapt these approaches for our setting by adding a contrastive vision-language objective and using zero-shot classification scores during fine-tuning; we refer to these approaches as VL-ERM and VL-GDRO respectively.

Table 3 summarizes mean zero-shot classification results across our real-world evaluation settings. Since performance of mitigation methods is dependent on the accuracy of the discovered spurious correlations in Stage 1, Table 3 displays results for two evaluation categories: (i) the 192 settings where RAVL Stage 1 Precision@10 is greater than 0.6, and (ii) the 106 settings where RAVL Stage 1 Precision@10 is greater than 0.8. In line with prior works on robustness [39, 56], we report image overall performance and image worst-group performance. Additionally, in order to evaluate the extent to which the VLM understands fine-grained features, we introduce two new metrics: region overall performance and region worst-group performance. Region-level accuracies are computed by performing zero-shot classification with region embeddings and comparing predicted labels to the ground-truth region-level labels provided in the zero-shot classification dataset.

Results show that RAVL consistently outperforms prior approaches in mitigating spurious correlations. Across the two evaluation categories in Table 3, RAVL contributes to an improvement of up to 8.2% on image worst-group performance and 10.8% on region worst-group performance over the nearest baseline. Improvements in region worst-group performance are particularly notable, suggesting that RAVL can better interpret fine-grained features when compared to prior approaches. Additionally, as the accuracy of the discovered spurious correlations in Stage 1 increases, the performance of the RAVL mitigation approach increases proportionally. Our results demonstrate the efficacy of our fine-tuning procedure in mitigating spurious correlations when compared to prior approaches.

## 5  Conclusion

In this work, we introduced RAVL, a fine-grained region-aware approach for addressing spurious correlations in VLMs. We demonstrate through large-scale, controlled experiments as well as in-the-wild evaluations that RAVL can discover (191% improvement in identified correlations) and mitigate (8.2% improvement on worst-group performance) spurious correlations in VLMs. We hope that our work can help (i) diagnose and correct critical failure modes in VLMs prior to deployment and (ii) drive progress towards the development of fine-grained approaches for model robustness.

## Acknowledgments and Disclosure of Funding

We are thankful to Sophie Ostmeier, Eduardo Reis, and Ashwin Kumar for helpful discussions and feedback. MV is supported by graduate fellowship awards from the Department of Defense (NDSEG), the Knight-Hennessy Scholars program at Stanford University, and the Quad program. AC is supported by NIH grants R01 HL167974, R01HL169345, R01 AR077604, R01 EB002524, R01 AR079431, P41 EB027060, AY2AX000045, and 1AYSAX0000024-01; and NIH contracts 75N92020C00008 and 75N92020C00021. CL is supported by NIH grants R01 HL155410, R01 HL157235, by AHRQ grant R18HS026886, and by the Gordon and Betty Moore Foundation. JBD and CL are supported by the Medical Imaging and Data Resource Center (MIDRC), which is funded by the National Institute of Biomedical Imaging and Bioengineering (NIBIB) under contract 75N92020C00021 and through The Advanced Research Projects Agency for Health (ARPA-H).

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

# Appendix

## Contents

## A  Related Work

Machine learning models often learn spurious correlations (also known as shortcuts) between image features and class labels. For instance, models have been shown to rely on the presence of chest tubes rather than disease features when identifying collapsed lungs in chest X-rays [34]; surgical skin markings when detecting melanoma from skin lesions [48]; and environmental features when performing object recognition tasks [3]. Models that learn spurious correlations will generalize poorly to real-world settings. Our work builds on several recent research directions for (i) discovering and (ii) mitigating spurious correlations.

**Discovering Spurious Correlations**. In the unimodal setting, prior works have developed automated methods for identifying systematic errors resulting from learned spurious correlations in vision models. Using a labeled validation set, these approaches utilize clustering algorithms [13, 43] or lightweight models [42, 22, 31] to identify subgroups of images with high error rates; for instance, in the example in Figure 1, images containing butterflies without flowers may be identified as one such subgroup. Given a set of images in the identified subgroups, a user can then identify the common features and rectify the data or model. However, recent work has suggested that it is often challenging for humans to interpret identified subgroups and accurately determine the shared features resulting in model failure [25]. Additionally, such methods often focus solely on identifying images with high error rates (e.g. butterflies without flowers) rather than identifying the specific class of features contributing to the error (e.g. flowers). A related line of work has aimed to identify spurious features using human supervision [41] or external concept banks [50].

In the vision-language setting, Yang et al. use an external off-the-shelf object detector to annotate features [56]. Then, for each feature, the difference in zero-shot classification accuracy between images containing the feature and those without the feature is measured; high performance gaps are used to signal spurious features. However, the efficacy of this approach is reliant on the quality of the object detector and a human-in-the-loop is used to verify results; also, as we show in this work, performance gaps alone are not always sufficient for discovering spurious features.

**Mitigating Spurious Correlations**. There is a line of work aiming to mitigate spurious correlations in the context of deep learning [61, 39, 32, 9, 28, 33, 21]. These works explore strategies like data augmentation [55, 59, 57, 22, 50] and instance upsampling [39, 43]. While these approaches have been explored widely in unimodal tasks [29, 53], mitigating spurious correlations in vision-language settings has not been extensively studied. Some previous works have studied this problem within the context of pretrained VLMs [60, 49, 1]; however, their setting differs markedly from the fine-tuned

VLM setting, where datasets are composed of image-text pairs with no class or subgroup labels. In the fine-tuned VLM setting, existing works predominantly operate at the global image-level [56], which is unlikely to be sufficient for mitigating fine-grained spurious correlations.

## B    Extended Details on Evaluation Settings

We create 654 evaluation settings using data from two domains: (1) synthetic data (MNIST [11] and FashionMNIST [51]) and (2) real-world data (COCO [27]). Below, we provide implementation details for the four components included in each evaluation setting:

1. *Predefined spurious correlation*: We define a spurious image feature and textual attribute pair ($\mathbf{e}^{eval}$, $a^{eval}$). For MNIST and FashionMNIST, $\mathbf{e}^{eval}$ represents a red rectangle; $a^{eval}$ is generated from the set {zero, one, two, three, four five, six, seven, eight, nine} for MNIST and {t-shirt, trouser, pullover, dress, coat, sandal, shirt, sneaker, bag, ankle boot} for FashionMNIST. For COCO, we sample $\mathbf{e}^{eval}$ and $a^{eval}$ from the list of annotated attributes.

2. *Fine-tuning dataset*: Vision-language fine-tuning datasets $\mathcal{D}_F^{eval}$ are sampled from the training sets of MNIST, FashionMNIST, and COCO such that the presence of feature $\mathbf{e}^{eval}$ is correlated with the presence of text attribute $a^{eval}$ as measured by Cramer's V. For MNIST and FashionMNIST, we synthetically generate text captions by randomly sampling from the following pre-defined prompt templates: THE IMAGE SHOWS A [CLASS LABEL], THE DIGIT APPEARS TO BE [CLASS LABEL], THERE IS AN IMAGE SHOWING A [CLASS LABEL], and THE NUMBER IS A [CLASS LABEL]. In order to reflect real-world settings where spurious features (e.g. skin markings in dermoscopic images [48]) may not be annotated in text, text captions in our synthetic settings solely refer to class labels and do not describe the spurious feature. For COCO, we use the provided text captions.

3. *Fine-tuned VLM*: We fine-tune each model $\mathcal{M}$ on dataset $\mathcal{D}_F^{eval}$ using a single NVIDIA A100 GPU with an initial learning rate of 5e-5. We use a batch size of 128 and train for 100 epochs with early stopping. We set the loss temperature as $\tau = 0.07$. In line with prior works that explore the benefits of locked image-text training [2, 46], we freeze the text encoder and only learn weights for the image encoder.

4. *Evaluation dataset*: We construct zero-shot classification datasets $\mathcal{D}_V^{eval}$ from the test sets of MNIST, FashionMNIST, and COCO. For MNIST and FashionMNIST, we generate region bounding boxes using equally-sized quadrants. For COCO, we use the ground-truth bounding boxes and associated labels. Evaluation datasets are sampled to ensure that a correlation between $a^{eval}$ and $\mathbf{e}^{eval}$ does not exist. For MNIST, we perform prompt ensembling for zero-shot classification using the following prompts: A PHOTO OF THE NUMBER [CLASS LABEL]; THE DIGIT [CLASS LABEL]; AN IMAGE OF A [CLASS LABEL]; [CLASS LABEL]. For FashionMNIST, we use the following prompts: A PHOTO OF A [CLASS LABEL]; THE [CLASS LABEL]; AN IMAGE OF A [CLASS LABEL]; [CLASS LABEL]. For COCO, we use the following prompts: THERE IS A [CLASS LABEL]; A PHOTO OF THE [CLASS LABEL]; A PHOTO OF A [CLASS LABEL]; [CLASS LABEL].

Our 654 evaluation settings are summarized in Table 4. Datasets are licensed under CC BY, CC BY-SA, CC BY-NC, or MIT licenses. In Figure 4, we provide examples of both synthetic and real-world evaluation settings.

Table 4: *Evaluation settings.* We evaluate our approach on 654 settings, divided across 2 data domains and 2 model initializations.

| Domain | Model $\mathcal{M}$ Initialization | |
| --- | --- | --- |
| | CLIP-RN50 | CLIP-RN101 |
| Synthetic Data | 129 | 162 |
| Real-World Data | 171 | 192 |

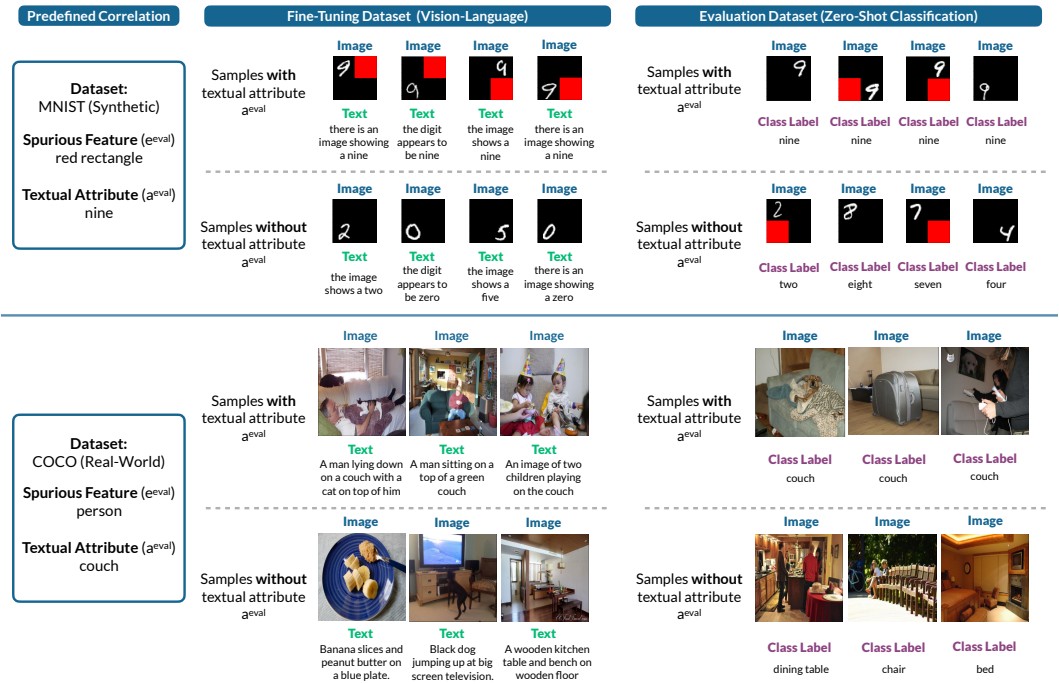

Figure 4: *Example evaluation settings.* Here, we provide examples of predefined spurious correlations, fine-tuning datasets, and evaluation datasets associated with a synthetic evaluation setting (top row) and a real-world evaluation setting (bottom row). The example synthetic evaluation setting consists of a predefined spurious correlation between a red rectangle (spurious image feature $\mathbf{e}^{eval}$) and `nine` (textual attribute $a^{eval}$). This spurious correlation is visible in the vision-language fine-tuning dataset, where the presence of red rectangles and nines are strongly correlated, but not in the evaluation dataset. Similarly, the example real-world evaluation setting consists of a predefined spurious correlation between a person (spurious image feature $\mathbf{e}^{eval}$) and `couch` (textual attribute $a^{eval}$). Again, this spurious correlation is visible in the vision-language fine-tuning dataset, where the presence of people and couches are strongly correlated, but not in the evaluation dataset.

## C   Extended Details on RAVL Mitigation

In this section, we extend Section 4.1 by providing additional descriptions of our region-aware loss function.

For batch $\mathcal{B}$, we define $\mathcal{R}_{\mathcal{B}}^s$ as the set of all spurious regions in the batch: $\mathcal{R}_{\mathcal{B}}^s = \bigcup_{I_i \in B} \mathcal{R}_i^s$. For image $I_i$ in batch $\mathcal{B}$, the first component of our region-aware loss function $L_R^i$ is designed to maximize embedding similarity between non-spurious regions $\mathcal{R}_i^r$ and assigned class label $\hat{y}_i$; simultaneously, $L_R^i$ will minimize embedding similarity between non-spurious regions $\mathcal{R}_i^r$ and other class labels in the batch. We formulate $L_R^i$ as follows:

$$L_R^i = -\log \frac{\sigma_m(\mathcal{R}_i^r, \hat{y}_i)}{\sum_{\hat{y}_j \in \mathcal{B}} \sigma_m(\mathcal{R}_i^r, \hat{y}_j) + P(\mathcal{R}_{\mathcal{B}}^s)}, \tag{5}$$

where $P(\mathcal{R}_{\mathcal{B}}^s)$ is a penalty term that encourages dissimilarity between spurious features and correlated class labels as expressed below. Including this term in the denominator of $L_R^i$ is meant to pull embeddings of spurious regions away from correlated class labels.

$$P(\mathcal{R}_{\mathcal{B}}^s) = \sum_{r_j \in \mathcal{R}_{\mathcal{B}}^s} \max_{\hat{y}_k \in \mathcal{B}} \sigma(r_j, \hat{y}_k) \tag{6}$$

The formula for $L_R^i$ includes two similarity functions: $\sigma$ and $\sigma_m$. We define $\sigma$ and $\sigma_m$ as follows. Let $f$ represent a region embedding function (associated with the image encoder of VLM $\mathcal{M}$) and

let $g$ represent a text embedding function (associated with the text encoder of VLM $\mathcal{M}$). Then, for an arbitrary region $a$, the function $f(a)$ will generate region embedding $f(a) \in \mathbb{R}^d$ with embedding dimension $d$. For an arbitrary class label $b$, $g(b)$ will generate text embedding $g(b) \in \mathbb{R}^d$. Given this notation, we establish the following definitions for $\sigma$ and $\sigma_m$:

$$\sigma(a, b) = \exp(\langle f(a), g(b) \rangle / \tau) \tag{7}$$

$$\sigma_m(\mathcal{A}, b) = \exp(\max_{a \in \mathcal{A}}(\langle f(a), g(b) \rangle / \tau)) \tag{8}$$

In the loss term $L_R^i$, the function $\sigma_m(\mathcal{R}_i^r, \hat{y}_i)$ will compute the maximum similarity between regions in $\mathcal{R}_i^r$ and class label $\hat{y}_i$. We specifically use the maximum operation in this computation since there are likely to be regions included in $\mathcal{R}_i^r$ that do not reflect the class label; for instance, in the example provided in Figure 1, there may be non-spurious regions such as trees or leaves included in $\mathcal{R}_i^r$, which do not align with the animal class labels. The maximum operation ensures that the similarity between *at least one* region in $\mathcal{R}_i^r$ and the class label should be high.

The second component of our region-aware loss function $L_A^i$ is designed to maximize embedding similarity between non-spurious regions $\mathcal{R}_i^r$ and assigned class label $\hat{y}_i$; simultaneously, $L_A^i$ will minimize embedding similarity between other regions in the batch and class label $\hat{y}_i$. We formulate $L_A^i$ as follows:

$$L_A^i = -\log \frac{\sigma_m(\mathcal{R}_i^r, \hat{y}_i)}{\sigma_m(\mathcal{R}_i^r, \hat{y}_i) + \sum_{j=1, \hat{y}_j \neq \hat{y}_i}^{|\mathcal{B}|} \sigma_m(\mathcal{R}_j^r, \hat{y}_i) + \sum_{r_j \in \mathcal{R}_\mathcal{B}^s} \sigma(r_j, \hat{y}_i)}. \tag{9}$$

# D   Extended Evaluations

## D.1   Extended Results for RAVL Stage 1 (Discovery)

In this section, we extend the results provided in Section 3.3 with additional evaluations of Stage 1 of RAVL. Our goal is to evaluate the ability of RAVL to discover fine-grained spurious correlations between image features and textual attributes.

### D.1.1   Extended Comparisons to Prior Approaches

We implement RAVL according to the details provided in Section 3.1. RAVL includes a clustering step that identifies groups of visually-similar regions. For all evaluation settings, we identify the optimal number of clusters by sweeping all cluster sizes ranging between $|\mathcal{Y}| * 2$ and $|\mathcal{Y}| * 5$; we then select the optimal number of clusters using Silhouette scores. We select these bounds to be larger than the class label set size by several multiples in order to ensure that clusters adequately separate distinct features. Prior works [13, 43] have also utilized overclustering approaches for this objective. We note that users can adjust the bounds based on the composition of their dataset; for instance, complex datasets with diverse features may require a larger range. For MNIST and FashionMNIST, the size of the label set $|\mathcal{Y}|$ is 10; For COCO, the size of the label set ranges between 2 and 5.

For all baselines, we utilize the official implementations provided by the authors. We adapt Domino, George, and Distilling Failures for our setting by providing region embeddings as input rather than image embeddings.

In Figure 5, we provided an extended version of Figure 2. We demonstrate that RAVL consistently outperforms baselines across two domains (synthetic images and real images), two model initializations (CLIP-RN50 and CLIP-RN101), and four learned correlation strengths (measured by varying $\tau_{eval} \in \{10, 20, 30, 40\}$).

### D.1.2   Additional details for in-the-wild evaluations

**Evaluations on scene classification**: Here, we provided extended details on our in-the-wild evaluations performed on scene images (Section 3.3).

We leverage ten off-the-shelf VLMs as our model $\mathcal{M}$: CLIP-RN50, OpenCLIP-RN50, CLIP-RN101, OpenCLIP-RN101, CLIP-ViTB/32, OpenCLIP-ViTB/32, CLIP-ViTB/16, OpenCLIP-ViTB/16, CLIP-ViTL/14, and OpenCLIP-ViTL/14 [36, 20]. The four RN models utilize ResNet vision encoders

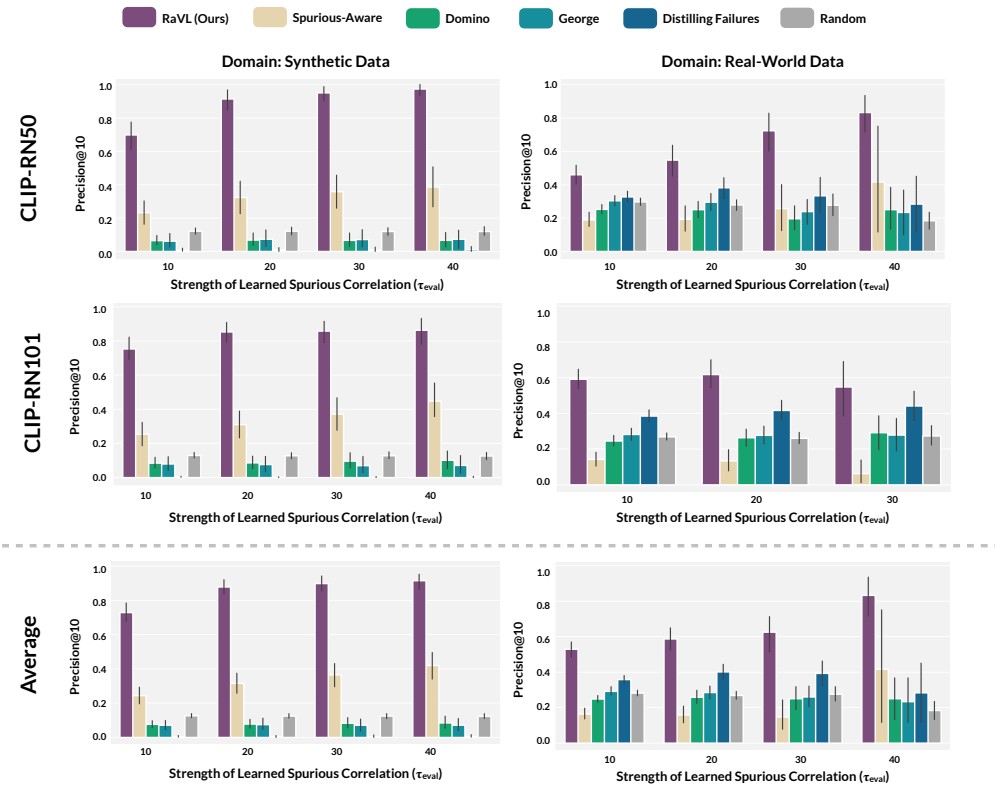

Figure 5: RAVL *accurately identifies spurious correlations.* Here, we provide an extended version of Figure 2, which demonstrates that RAVL consistently outperforms prior methods in discovering learned spurious correlations between image features and textual attributes. Here, we provide Precision@10 metrics for a CLIP-RN50 model fine-tuned on synthetic data (129 settings) and real-world data (171 settings); a CLIP-RN101 model fine-tuned on synthetic data (162 settings) and real-world data (192 settings); and an average across both model architectures.

and the six ViT models utilize Vision Transformer backbones. The CLIP models were trained on a proprietary dataset with 400M image-text pairs. OpenCLIP ResNet models were trained on YFCC15M [44], and OpenCLIP ViT models were trained on LAION2B [40].

We select SUN397 as our zero-shot classification dataset $\mathcal{D}_V$ [52]. SUN397 consists of scene images from 397 classes. We use the test data from official partition number 1, which consists 19,850 images. We then use an off-the-shelf region proposal network [63] to identify candidate regions.

For each VLM $\mathcal{M}$, we perform 397-class zero-shot scene classification on SUN397. We use a prompt ensemble consisting of two prompt templates as provided by CLIP [36]. Due to the large size of the zero-shot classification dataset $\mathcal{D}_V$, we perform clustering using the CLARA (Clustering for Large Applications) algorithm, which is an efficient implementation of K-Medoids, and fix the number of clusters as $|\mathcal{Y}| * 2$, which is 794 in this case.

**Evaluations on chest X-ray classification**: Here, we provided extended details on our in-the-wild evaluations performed on medical images (Section 3.3). In recent years, a range of vision [47, 54, 37] and vision-language [45, 5, 4, 62] models have been proposed for learning diagnostic patterns in medical images, and there is a critical need for methods capable of identifying spurious correlations in this domain. Our goal is to determine if RAVL can effectively surface spurious correlations learned by real-world fine-tuned VLMs developed for medical image interpretation.

We leverage two off-the-shelf variants of the PubMedCLIP model [12] as our VLM $\mathcal{M}$: PubMedCLIP-RN50 and PubMedCLIP-ViTB/32. The PubMedCLIP-RN50 model utilizes a ResNet-50 vision encoder and was fine-tuned from the CLIP-RN50 model. The PubMedCLIP-ViTB/32 model utilizes a Vision Transformer backbone for the vision encoder and was fine-tuned from the CLIP-ViTB/32

model. Both variants of PubMedCLIP are fine-tuned using ROCO, a large radiology dataset consisting of images and captions collected from PubMed [35].

We select Object-CXR as our zero-shot classification dataset $\mathcal{D}_V$. Object-CXR is a dataset of 10,000 frontal chest X-rays compiled from around 300 township hospitals in China [23]. Twelve radiologists with 1-3 years of experience annotated the images, identifying foreign objects within the lung field using bounding boxes, ellipses, or masks, excluding support devices. We retain only bounding boxes and exclude chest X-rays without annotations, resulting in 8,726 object annotations across 4,372 images. For our evaluations, we use the Object-CXR dev split, which includes 974 object annotations across 489 images. We assign image-level labels to the dataset using torchxrayvision [8], a library that includes a variety of pretrained chest X-ray models. Specifically, we use the XRV-DENSENET121-DENSENET121-RES224-ALL pretrained model to produce multi-class labels for a variety of diseases, including Enlarged Cardiomediastinum, Cardiomegaly, Lung Opacity, Lung Lesion, Edema, Consolidation, Pneumonia, Atelectasis, Pneumothorax, Pleural Effusion, and Fracture. A disease is identified as present if it meets a confidence threshold of 0.60.

For each PubMedCLIP VLM $\mathcal{M}$, we perform binary zero-shot classification of cardiomegaly in Object-CXR. Cardiomegaly is a medical condition characterized by the presence of an enlarged heart. After performing a manual search over the text prompt space, we identify CARDIOMEGALY and NORMAL as the prompts that lead to the highest zero-shot classification accuracy for both model variants. The PubMedCLIP-RN50 model achieves an overall zero-shot classification accuracy of 74.2, with an accuracy of 14.0 on the group of images with cardiomegaly and an accuracy of 91.9 on the group of images without cardiomegaly. The PubMedCLIP-ViTB/32 achieves an overall zero-shot classification accuracy of 39.4, with an accuracy of 80.4 on the group of images with cardiomegaly and an accuracy of 28.0 on the group of images without cardiomegaly. Interestingly, given the selected prompts, we note that the PubMedCLIP-RN50 and the PubMedCLIP-ViTB/32 models exhibit inverse trends, with PubMedCLIP-RN50 achieving higher performance on the class of images without cardiomegaly and PubMedCLIP-ViTB/32 achieving higher performance on the class of images with cardiomegaly.

Given VLM $\mathcal{M}$ and zero-shot classification dataset $\mathcal{D}_V^{eval}$, we apply RAVL in order to surface learned spurious correlations. Similar to our controlled evaluations on synthetic datasets, we perform K-Medoids clustering with the number of clusters ranging from 20 to 50. The optimal number of clusters is selected using Silhouette distance; we use 24 clusters for PubMedCLIP-RN50 and 20 clusters for PubMedCLIP-ViTB/32. The final cluster performance gap metric $G_c$ associated with the top-ranked spurious feature cluster is 0.041 and 0.119 for the PubMedCLIP-RN50 and PubMedCLIP-ViTB/32 models respectively.

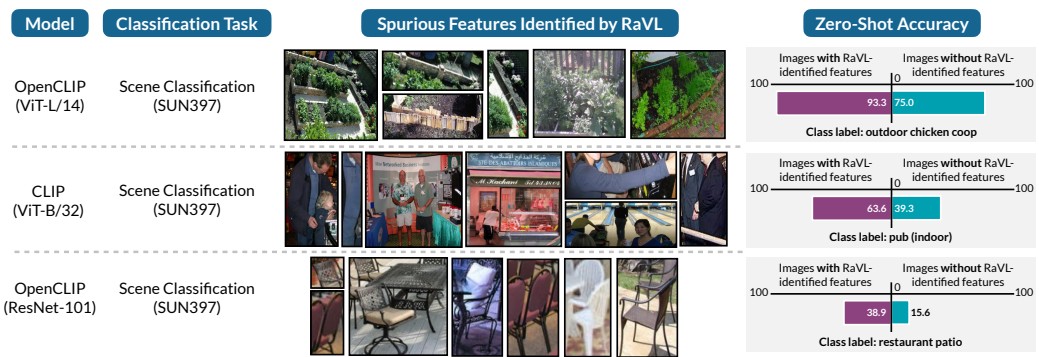

Figure 6: RAVL *surfaces spurious correlations in off-the-shelf VLMs.* Here, we extend Figure 3 with additional examples of spurious correlations discovered by RAVL in off-the-shelf-VLMs.

**Extended Results:** In Figure 6, we extend the qualitative results provided in Figure 3 with additional examples of spurious correlations surfaced by RAVL in off-the-shelf VLMs. We make the following observations:

- For the OpenCLIP ViT-L/14 model, RaVL surfaces a feature cluster consisting of green plants and fences. We observe a performance gap of 18.3 points between images with class

label `outdoor chicken coop` that contain the RaVL-identified feature and those that do not contain the feature. This suggests that the OpenCLIP ViT-L/14 model can better classify an `outdoor chicken coop` scene when green plants and fences are present.

- For the CLIP ViT-B/32 model, RaVL surfaces a feature cluster consisting of people. We observe a performance gap of 24.3 points between images with class label `pub (indoor)` that contain the RaVL-identified feature and those that do not contain the feature. This suggests that the CLIP ViT-B/32 model can better classify a `pub (indoor)` scene when people are present.

- For the OpenCLIP ResNet-101 model, RaVL surfaces a feature cluster consisting of chairs. We observe a performance gap of 23.3 points between images with class label `restaurant patio` that contain the RaVL-identified feature and those that do not contain the feature. This suggests that the OpenCLIP ResNet-101 model can better classify `restaurant patio` scenes when chairs are present.

### D.2 Extended Results for RAVL Stage 2 (Mitigation)

We train model $\mathcal{M}_{new}$ using a single NVIDIA A100 GPU with an initial learning rate of 5e-5. We use a batch size of 128 and train for 100 epochs with early stopping. We set the loss temperature as $\tau = 0.07$ and use $\lambda = 0.8$ in loss function $\mathcal{L}$. In line with prior works that utilize locked image-text fine-tuning [2, 46], we freeze the text encoder and solely learn weights for the image encoder. We generate candidate regions for the fine-tuning dataset $\mathcal{D}_F^{eval}$ using a region proposal network with identical settings to prior work [63].

Below, we provide additional implementation details for the five mitigation baselines we explore in this study. Since there are a limited number of existing approaches designed for mitigating spurious correlations in fine-tuned VLMs, we adapt several existing methods for our setting in order to train model $\mathcal{M}_{new}$:

- *Standard VLM Fine-Tuning:* We perform standard VLM fine-tuning with the original loss function $L_{CL}$ used to train model $\mathcal{M}$. In our experiments, $L_{CL}$ is the CLIP objective [36].

- *Upsampled VLM Fine-Tuning:* We perform VLM fine-tuning with the original loss function $L_{CL}$ used to train model $\mathcal{M}$. In our experiments, $L_{CL}$ is the CLIP objective [36]. We utilize a weighted sampler to upsample minority groups during training; class and subgroup labels are derived from Stage 1 of RAVL as detailed in Section 4.1.

- *VL-ERM:* Since empirical risk minimization (ERM) is traditionally used in unimodal classification settings, we adapt ERM for our multimodal setting by incorporating an extra contrastive vision-language objective function; this loss function is intended to ensure that VLM $\mathcal{M}_{new}$ learns image-text relationships during training. Specifically, the final loss function during training is $\mathcal{L}_{VLERM} = \lambda L_{CL} + (1 - \lambda)L_{ERM}$. Here, $L_{CL}$ takes the form of the original loss function used to train model $\mathcal{M}$; in our experiments $L_{CL}$ is the CLIP objective [36]. We set $\lambda = 0.8$. During training, we apply ERM to zero-shot classification logits computed using image embeddings and text embeddings of class labels. We utilize a weighted sampler to upsample minority subgroups; class and subgroup labels are derived from Stage 1 of RAVL as detailed in Section 4.1.

- *VL-GDRO* [39]: Since GDRO is traditionally used in unimodal classification settings, we adapt GDRO for our multimodal setting by incorporating an extra contrastive vision-language objective function; this loss function is intended to ensure that VLM $\mathcal{M}_{new}$ learns image-text relationships during training. Specifically, the final loss function during training is $\mathcal{L}_{VLGDRO} = \lambda L_{CL} + (1 - \lambda)L_{GDRO}$. Here, $L_{CL}$ takes the form of the original loss function used to train model $\mathcal{M}$; in our experiments $L_{CL}$ is the CLIP objective [36]. We set $\lambda = 0.8$. During training, we apply GDRO to zero-shot classification logits computed using image embeddings and text embeddings of class labels. In line with standard practice, we utilize a weighted sampler to upsample minority subgroups; class and subgroup labels are derived from Stage 1 of RAVL as detailed in Section 4.1.

- *Spurious-Aware Mitigation* [56]: Spurious-aware mitigation aims to address spurious correlations in VLMs using a combination of five loss functions: one CLIP objective function, two contrastive image objective functions meant to address spurious correlations in the image space, and two contrastive language objective functions meant to address spurious

Table 5: RAVL *effectively mitigates spurious correlations across various model initializations.* Here, we provide an extended version of Table 3 with a breakdown of results by model initialization (CLIP-RN50 vs. CLIP-RN101). Our results demonstrate that RAVL consistently outperforms prior methods in mitigating spurious correlations. We report mean Image Overall (Img. Overall), Image Worst Group (Img. WG), Region Overall (Reg. Overall), and Region Worst Group (Reg. WG) metrics across our real-world evaluation settings.

| | Method | Discovery Precision@10$> 0.6$ | | | | Discovery Precision@10$> 0.8$ | | | |
|---|---|---|---|---|---|---|---|---|---|
| | | Img. Overall | Img. WG | Reg. Overall | Reg. WG | Img. Overall | Img. WG | Reg. Overall | Reg. WG |
| CLIP-RN50 | Standard FT | 64.2 | 35.8 | 73.2 | 50.1 | 64.4 | 36.0 | 74.3 | 51.8 |
| | Upsampled FT | 65.2 | 36.7 | 73.7 | 51.0 | 65.2 | 38.0 | 74.3 | 52.3 |
| | VL-ERM | 66.0 | 30.0 | 73.4 | 45.2 | 65.4 | 28.9 | 73.7 | 45.4 |
| | VL-GDRO | 66.8 | 31.6 | 74.1 | 45.8 | 66.1 | 29.6 | 74.7 | 46.3 |
| | Spurious-Aware | 67.4 | 31.6 | 74.0 | 45.2 | 66.2 | 28.2 | 74.5 | 45.2 |
| | RAVL (Ours) | **67.9** | **36.9** | **77.8** | **55.4** | **67.8** | **38.6** | **79.0** | **56.5** |
| CLIP-RN101 | Standard FT | 63.9 | 28.9 | 71.3 | 45.0 | 64.7 | 27.6 | 72.0 | 44.4 |
| | Upsampled FT | 67.4 | 38.4 | 74.6 | 52.8 | 67.8 | 37.5 | 75.0 | 53.2 |
| | VL-ERM | 70.5 | 33.5 | 77.0 | 53.4 | 70.8 | 32.2 | 77.4 | 54.0 |
| | VL-GDRO | 70.5 | 34.9 | 76.5 | 53.1 | 70.5 | 32.1 | 76.8 | 54.1 |
| | Spurious-Aware | **71.3** | 34.8 | 78.1 | 53.9 | 71.2 | 32.4 | 78.4 | 53.9 |
| | RAVL (Ours) | 71.0 | **40.4** | **79.5** | **59.2** | **71.8** | **42.2** | **79.8** | **59.8** |

correlations in the text space. We note that Spurious-Aware Mitigation was explicitly designed for the fine-tuned VLM setting. We follow the implementation of Spurious-Aware Mitigation provided by [56]. In our work, since we solely fine-tuned the vision encoders of VLMs $\mathcal{M}$, we use a version of Spurious-Aware Mitigation with the CLIP objective function and two contrastive image objective functions.

Prior works on model robustness predominantly evaluate model performance using image worst-group scores [39]. In addition to image worst-group accuracy, we also report region overall and region worst-group accuracies, which evaluate the extent to which the VLM understands fine-grained features. Region-level accuracies are computed by performing zero-shot classification with region embeddings and comparing predicted labels to the ground-truth region-level labels provided in the zero-shot classification dataset.

In Table 5, we provide an extended version of Table 3 with a breakdown of results by model initialization (CLIP-RN50 and CLIP-RN101). We demonstrate that RAVL consistently outperforms prior methods across both model initializations. In Table 6, we provide an extended version of Table 3 with a breakdown of results by the learned correlation strength of the original VLM $\mathcal{M}$. RAVL consistently outperforms prior methods across four correlation strengths $\tau_{eval} \in \{10, 20, 30, 40\}$.

### D.3 Computational Complexity Analysis

In this section, we provide an analysis of the computational complexity of RAVL. RAVL is computationally inexpensive; in particular, the RAVL discovery stage can be run efficiently on CPU and the mitigation stage adds only a small computational overhead. Below, we provide an analysis of computational complexity for each stage of RAVL.

**Computational complexity analysis of RAVL Stage 1:** The discovery stage of RAVL is specifically designed to be run on a labeled validation dataset $\mathcal{D}_V$; in real-world settings, validation datasets are often relatively small in size due to the human effort needed for securing labels, rendering this stage as computationally inexpensive for diverse applications. Even if the validation dataset is large in size, RAVL operates efficiently as follows:

- First, RAVL preprocesses images by decomposing each image into candidate regions; there are a variety of ways in which a user can decompose an image into regions, such as by using equal-sized segments (e.g. quadrants) or running inference with region proposal networks (RPNs). Both methods are inexpensive and only need to be run once in an offline manner. Similar approaches have been applied to large-scale datasets in prior work [63].

- Then, embeddings need to be generated for each region, which can be done by utilizing VLM $\mathcal{M}$ for inference (forward passes only). Across a set of 10 FashionMNIST and COCO

Table 6: RAVL *effectively mitigates spurious correlations across learned correlation strengths.* Here, we provide an extended version of Table 3 with a breakdown of results by the learned correlation strength ($\tau_{eval} \in \{10, 20, 30, 40\}$ of the original model $\mathcal{M}$. We use the subset of 106 evaluation settings where RAVL Stage 1 Precision@10 is greater than 0.8. Our results demonstrate that RAVL consistently outperforms prior methods in mitigating spurious correlations across various correlation strengths and model initializations. We report mean Image Worst Group (Img. WG) and Region Worst Group (Reg. WG) metrics across our real-world evaluation settings. We note that there are no valid evaluation settings for CLIP-RN101 when the learned correlation strength $\tau_{eval}$ of the original model $\mathcal{M}$ is set to 40.

| | Method | $\tau_{eval} = 10$ | | $\tau_{eval} = 20$ | | $\tau_{eval} = 30$ | | $\tau_{eval} = 40$ | |
| | | Img. WG | Reg. WG | Img. WG | Reg. WG | Img. WG | Reg. WG | Img. WG | Reg. WG |
|---|---|---|---|---|---|---|---|---|---|
| CLIP-RN50 | Standard FT | 36.0 | 51.8 | 29.0 | 43.8 | 31.4 | 46.6 | 26.0 | 39.5 |
| | Upsampled FT | 38.0 | 52.3 | 27.7 | 43.5 | 33.4 | 47.3 | 33.4 | 46.5 |
| | VL-ERM | 28.9 | 45.4 | 23.3 | 38.8 | 25.7 | 38.3 | 22.9 | 32.5 |
| | VL-GDRO | 29.6 | 46.3 | 22.7 | 37.1 | 28.6 | 37.6 | 26.4 | 28.2 |
| | Spurious-Aware | 28.2 | 45.2 | 22.2 | 37.9 | 24.6 | 38.4 | 17.1 | 30.6 |
| | RAVL (Ours) | **38.6** | **56.5** | **35.3** | **56.8** | **41.0** | **60.2** | **38.0** | **53.5** |
| CLIP-RN101 | Standard FT | 27.6 | 44.4 | 26.4 | 43.1 | 17.3 | 35.1 | – | – |
| | Upsampled FT | 37.5 | 53.2 | 36.0 | 49.4 | 25.6 | 40.0 | – | – |
| | VL-ERM | 32.2 | 54.0 | 32.1 | 51.5 | 18.1 | 42.5 | – | – |
| | VL-GDRO | 32.1 | 54.1 | 30.7 | 52.8 | 16.6 | 45.3 | – | – |
| | Spurious-Aware | 32.4 | 53.9 | 30.4 | 51.2 | 19.3 | 48.3 | – | – |
| | RAVL (Ours) | **42.2** | **59.8** | **39.7** | **56.4** | **28.8** | **54.0** | – | – |

    evaluation settings, we observe embedding generation to take a mean of 24.5 seconds on a single A100 GPU.

- Finally, given candidate regions and corresponding embeddings, the remainder of the RaVL discovery procedure (clustering and computation of metrics) can be run completely on CPU. Across a set of 10 evaluation settings on COCO and FashionMNIST, we observe that clustering and computation of metrics require a mean of 3.4 seconds to run.

**Computational complexity analysis of RAVL Stage 2:** The mitigation stage of RAVL requires finetuning a VLM $\mathcal{M}_{new}$. Across a set of 10 evaluation settings on COCO and FashionMNIST, we observe that the inclusion of our fine-grained region-aware loss function at this stage adds an average of 0.15 seconds per training step (on a single A100 GPU) in comparison to the original fine-tuning procedure for $\mathcal{M}$.

# E   Extended Discussion

**Societal Impact:** The goal of our work is to improve robustness of fine-tuned VLMs to spurious correlations. As VLMs become more commonplace in society, we hope that our approach can enable users to better detect and mitigate model failures prior to deployment. We also note that our work includes a series of evaluations on medical images; rigorous clinical testing is necessary before robustness approaches are deployed in healthcare settings.

**Limitations:** In line with prior works in vision-only and vision-language settings, our method is specifically designed to surface and mitigate local, fine-grained spurious features. There may be some sources of spurious signal that do not manifest in this way; for instance, features like image brightness or gender can be considered global features, where the spurious signal is not localized to a particular image region. Our approach is not designed for these global spurious features. Rather, our problem setting is inspired by the many real-world, practical examples of region-level spurious features that have been demonstrated in literature to affect model performance, such as image-level markings in dermoscopic images [48], medical devices in radiographs [34], and text markers in medical images [10].

