# OpenReview forum: "RaVL: Discovering and Mitigating Spurious Correlations in Fine-Tuned Vision-Language Models"
_NeurIPS.cc/2024/Conference — NeurIPS 2024 poster_

### Official Review · Reviewer_onxK · 2024-07-12

**Soundness:** 3
**Presentation:** 3
**Contribution:** 3
**Rating:** 5
**Confidence:** 4

**Summary:**

This paper introduces RAVL, a method designed to identify and address spurious correlations in Vision-Language Models (VLMs). The authors emphasize two key areas: (a) prioritizing local image features over global image-level features, and (b) concentrating on the fine-tuning phase rather than the pre-training phase. They implement the first idea through the following steps: 1. Utilizing the VLM embedding space to extract candidate image features from the validation set. 2. Filtering these candidate features to pinpoint those that directly contribute to classification errors. 3. Ranking the remaining image features based on the extent of their learned spurious correlations. To mitigate spurious correlations, the authors regularize the dissimilarity between the spurious regions and correlated labels while encouraging high embedding similarity.

**Strengths:**

1. The paper is well-structured and easy to follow.

2. The proposed framework is sensible and, to the best of my knowledge, novel. I believe its contribution is slightly above the acceptance borderline.

3. The enhancement of performance quantitatively demonstrates the proposed method. It is notable.

**Weaknesses:**

1. This framework is limited to mitigating geometric spurious correlations by focusing on regional candidates. While some studies such as [1] highlight texture and background information as significant contributors to spurious correlations, it is more desirable to address all potential sources of spurious correlations.ss all the potential spurious correlations.

2. Although they used RoIAlign, there are no qualitative visualization results to show which regions affect which classes. I believe this would be important for understanding the model's behavior from a human vision perspective.

[1] Geirhos, Robert, et al. "ImageNet-trained CNNs are biased towards texture; increasing shape bias improves accuracy and robustness." arXiv preprint arXiv:1811.12231 (2018).

**Questions:**

1. What if this framework, instead of using RoIAlign, considers feature maps as candidates and filters them channel-wise? If this approach is effective, it could provide a more general framework from the perspective of Weakness 1, as feature maps are activated differently, representing various features [1].

2. Why did the authors use K-Medoids clustering? Additionally, I am curious about the authors' choice to use softmax-normalized embeddings. Feature maps might offer more diverse information for comparison during clustering.

3. Is this framework robust on batch size?


[1] Jeon, Myeongho, Myungjoo Kang, and Joonseok Lee. "A Unified Framework for Robustness on Diverse Sampling Errors." Proceedings of the IEEE/CVF International Conference on Computer Vision. 2023.

**Limitations:**

The authors discussed the limitations and potential negative societal impacts of their work in the Appendix. I fully agree with their assessment.

---

> ### Author Rebuttal · Authors · 2024-08-07
>
> We thank Reviewer ONXK for reviewing our work and providing helpful feedback.
>
> **[Q1] Assumption of region-level spurious correlations.**
>
> We thank the reviewer for raising this point. In line with prior works in vision-only and vision-language settings [1,2,3], RaVL is specifically designed to surface and mitigate fine-grained, region-level spurious features. As we acknowledge in the Appendix (Lines 779-787), we agree with the reviewer that there are some sources of spurious signal that do not manifest in this way (e.g. global, image-level features like texture or background information as mentioned in Geirhos et al.). However, we focus explicitly on region-level spurious features due to the fact that such spurious correlations have been shown to negatively affect model performance in many real-world, practical settings; some notable examples include image-level markings in dermoscopic images [8], medical devices in radiographs [9], and text markers in medical images [10]. As a result, we believe our work effectively complements work done by Geirhos et al. and others by targeting an important yet underresearched category of spurious correlations (namely, region-level features) that have been shown to contribute to significant model performance gaps.
>
> **[Q2] Visualizations.**
>
> Thank you for this suggestion. We refer the reviewer to the PDF provided with the general response. In Rebuttal Figure 2, we demonstrate the utility of our mitigation approach by providing qualitative visualizations using GradCAM. Visualizations show that after the RaVL mitigation procedure, the VLM $\mathcal{M}_{new}$ more precisely relies on core features.
>
> We also note that in order to quantitatively evaluate the extent to which the VLM understands fine-grained region-level information, we introduced the Region Overall and Region Worst Group metrics in Section 4.2. Results in Table 3 show that RaVL contributes to improvements on these metrics, suggesting better understanding of region-level information.
>
> **[Q3] Use of feature maps.**
>
> We agree that feature maps serve as a rich source of signal and can be leveraged for identifying and addressing spurious correlations. Indeed, several works have explored this direction by identifying neural features that influence classification decisions and then using humans in the loop to annotate feature maps as core or spurious [2,11]. Although this approach was shown to work well, a key consideration is the need for human supervision in order to interpret feature maps and distinguish between core and spurious features.
>
> In our work, we used ROIAlign and region-level embeddings rather than feature maps for the following reasons:
> - Enable automated analysis: Our approach is designed to operate without the need for human supervision, enabling generalizability to datasets from diverse domains (e.g. medical image data). Using region-level embeddings enables us to perform automated detection and mitigation of spurious correlations via our novel clustering approach and our region-aware loss function. On the other hand, using feature maps will likely require human supervision, as shown by [2] and [11].
> - Perform dimensionality reduction: Whereas feature maps may be large in size with many channels, region-level embeddings exhibit low dimensionality while still capturing relevant semantic information. With regards to the reviewer's point about the clustering stage, using softmax-normalized region-level embeddings can enable more efficient and accurate clustering in comparison with using large feature-maps that encompass diverse features.
> - Target fine-grained spurious correlations: As discussed in [Q1], our goal is to specifically detect and mitigate fine-grained, region-level spurious correlations. Using local region-level embeddings rather than global feature maps helps us accomplish this goal.
>
> **[Q4] Clustering approach.**
>
> We used K-medoids clustering for the following reasons:
> - Robustness to outliers: When compared to a standard K-Means clustering approach, K-Medoids is less sensitive to the presence of outliers due to the fact that cluster centroids are medians rather than means. Robustness to outliers is important in our setting, since region-level embeddings represent visually-diverse features and there is likely to be significant variation across samples.
> - Ability to operate with custom distance functions: Whereas K-Means traditionally minimizes the Euclidean distance between the centroid and the samples in a cluster, K-Medoids can be implemented with custom distance functions. Due to the fact that the original VLM $M$ is trained with respect to cosine distance, we implement K-Medoids with a cosine distance metric in this work in order to effectively model the patterns learned by $M$.
>
> **[Q5] Batch Size.**
>
> In order to train model $M_{new}$, we utilize a combination of the CLIP loss $L_{CL}$ and our region-aware loss $L_R + L_A$. Both loss functions are contrastive in nature and require a large set of negative samples in a batch in order to learn useful representations. As a result, our framework is sensitive to batch size; this is expected behavior [12] and is consistent with a long, existing line of work on contrastive learning methods (e.g. [7]). We note that for all results reported in Table 3, we keep batch size consistent across methods in order to enable fair comparisons. Below, we provide a brief batch size ablation on a single COCO evaluation setting. As expected, significant reductions in the batch size reduce efficacy of contrastive learning.
>
> | Batch Size | Method | Image Worst-Group |
> |----------------------|------------------|------------------|
> | 128 | Standard FT    | 51.5 |
> | 128 | RaVL (Ours) | **62.1** |
> | 32 | Standard FT | 39.4 |
> | 32 | RaVL (Ours) | **48.3** |
>
> We again thank Reviewer ONXK for their review of our manuscript and their positive overall assessment of our work. We hope that the above responses adequately address all concerns.

---

> > ### Comment · Reviewer_onxK · 2024-08-10
> > **Response to Author Feedback**
> >
> > Thank you for your feedback. However, after further consideration, I have become less favorable toward this submission and have decided to downgrade my initial score for the following reasons:
> >
> > - In response to Q1, the authors suggest that the empirical results are a key justification. However, with only a brief experiment provided in Q5, I am not convinced that RaVL outperforms other baselines across various batch sizes. The impact of batch size is not a trivial factor.
> >
> > - Thank you for including the additional qualitative experiment. However, I find the statement "The new VLM more precisely relies on core features" unclear. Could you please specify which core features you are referring to?
> >
> > - (i) Is it not possible to mitigate shortcut correlations (SCs) in an automated manner using feature maps? (ii) This makes sense. (iii) Why do you believe that feature maps represent global features? Given the receptive field, feature maps typically capture local features within the feature space (e.g., the first channel corresponds to the top-left, while the last channel corresponds to the bottom-right).
> >
> > - The justification provided for the use of K-medoids clustering seems appropriate to me.

---

> > > ### Comment · Reviewer_onxK · 2024-08-10
> > > **Response to Author Feedback**
> > >
> > > Please let me know if I misunderstood anything.

---

> > > > ### Author Response · Authors · 2024-08-14
> > > > **Response to Reviewer ONXK (1/2)**
> > > >
> > > > Below, we respond to the points raised by Reviewer ONXK during the discussion period.
> > > >
> > > > **Batch Size. “With only a brief experiment provided in Q5, I am not convinced that RaVL outperforms other baselines across various batch sizes. The impact of batch size is not a trivial factor.”**
> > > >
> > > > In our paper, we used a batch size of 128 across all mitigation experiments (Section 4). We selected a batch size of 128 for the following reasons. First, a batch size of 128 is the largest size that will fit on a single A100 GPU; we specifically used a single A100 GPU for all experiments in order to demonstrate the ease of use and cost-effectiveness of RaVL. Second, a batch size of 128 is either larger or equivalent to the batch sizes used by prior works that fine-tune VLMs for real-world applications (e.g. medical settings [1,2]) demonstrating that our choice of the batch size hyperparameter is consistent with real-world practice. Third, the Spurious-Aware Mitigation baseline, which is the only other previously-developed approach for the fine-tuned VLM setting (Appendix A), is designed to operate with a batch size of 128; consequently, using a batch size of 128 across evaluations enables a fair comparison with this baseline. As shown in Table 3 in the paper, RaVL Stage 2 outperforms all mitigation baselines.
> > > >
> > > > Beyond the batch size justification shared above, to be further responsive to the provided reviewer comment, we extend the batch size ablations provided with our original rebuttal by evaluating RaVL across three batch sizes (32, 128, and 256) and six baselines. We evaluate multiple COCO evaluation settings for each ablation. Across a total of 72 trained VLMs $M_{new}$ evaluated in the tables below, **we show that RaVL outperforms all baselines across all evaluated batch sizes**. We thank the reviewer for this suggestion, and we will be sure to update the manuscript with this ablation.
> > > >
> > > > | Batch Size | Method | Img. Overall 	| Img. Worst Group | Reg. Overall | Reg. Worst Group|
> > > > |----------------------|------------------|------------------|------------------|------------------|------------------|
> > > > | 32 | Standard FT   	| 59.1 	| 25.2	  | 69.3	 |44.1 	 |
> > > > | 32 | Upsampled FT	| 60.3 	| 28.0	 | 65.2 	| 34.3	|
> > > > | 32 | VL-ERM 		|58.4	 |21.6	  | 67.7	 | 37.2	|
> > > > | 32 | VL-GDRO 	 	| 59.6 	|24.3	 | 69.3	 | 35.2	|
> > > > | 32 | Spurious-Aware 	|54.2 	|12.2	 | 66.9	 | 36.2	|
> > > > | 32 | RaVL (Ours)		| **64.3**| **39.7**| **75.5** |**44.7**|
> > > >
> > > >
> > > > | Batch Size | Method | Img. Overall	 | Img. Worst Group | Reg. Overall | Reg. Worst Group|
> > > > |----------------------|------------------|------------------|------------------|------------------|------------------|
> > > > | 128 | Standard FT    	| 67.6 	| 33.6	 | 77.3 	| 46.0 	|
> > > > | 128 | Upsampled FT 	| 69.5	 | 37.1	 | 79.6 	| 52.9	 |
> > > > | 128 | VL-ERM		 | 68.2 	| 29.0	| 78.1	| 46.3	|
> > > > | 128 | VL-GDRO 		 | 69.7	 | 31.5	 | 77.5 	| 44.6	|
> > > > | 128 | Spurious-Aware 	| 69.4	 | 31.8	 | 78.6 	| 48.1	|
> > > > | 128 | RaVL (Ours) 	| **71.0** | **42.9** | **82.1** | **57.4**|
> > > >
> > > >
> > > > | Batch Size | Method | Img. Overall 	| Img. Worst Group | Reg. Overall | Reg. Worst Group|
> > > > |----------------------|------------------|------------------|------------------|------------------|------------------|
> > > > | 256 | Standard FT    	| 64.3 	| 41.4	 | 73.1 	| 53.9 	|
> > > > | 256 | Upsampled FT 	| 66.8	 | 48.8	 | 74.0 	| 58.5	 |
> > > > | 256 | VL-ERM		 | 62.5 	| 24.9	| 73.2	| 44.9	|
> > > > | 256 | VL-GDRO 		 | 63.5	 | 29.8	 | 75.2 	| 45.1	|
> > > > | 256 | Spurious-Aware 	| 66.7	 | 34.3	 | 74.6 	| 44.5	|
> > > > | 256 | RaVL (Ours) 	| **68.1** | **53.9** | **79.0** | **66.7**|
> > > >
> > > > **Visualizations. “Thank you for including the additional qualitative experiment. However, I find the statement "The new VLM more precisely relies on core features" unclear. Could you please specify which core features you are referring to?”**
> > > >
> > > > The images in the figure are obtained from the test set of a COCO evaluation setting in which the fine-tuned VLM $\mathcal{M}$ learned a strong spurious correlation between the class label "chairs" and the spurious feature "person". Visualizations show that before mitigation, the VLM is unable to precisely localize the core features associated with the class labels, often relying on the presence of people or various background details. After the RaVL mitigation procedure, the VLM $\mathcal{M}_{new}$ more precisely relies on core features (namely, the chairs).

---

> > > > > ### Author Response · Authors · 2024-08-14
> > > > > **Response to Reviewer ONXK (2/2)**
> > > > >
> > > > > **Feature Maps. “(i) Is it not possible to mitigate shortcut correlations (SCs) in an automated manner using feature maps? (ii) This makes sense. (iii) Why do you believe that feature maps represent global features? Given the receptive field, feature maps typically capture local features within the feature space (e.g., the first channel corresponds to the top-left, while the last channel corresponds to the bottom-right).”**
> > > > >
> > > > > Our approach uses RoIAlign to extract relevant regions of the feature maps, which we then compress into a low-dimensional embedding. These embeddings (1) can then be accurately and efficiently processed in an automated fashion and (2) are explicitly designed to characterize local, region-level features. We emphasize that using RoIAlign to embed regions of interest is a standard technique grounded in prior literature (e.g. [3]).
> > > > >
> > > > > (i) Performing automated identification of region-level spurious correlations using raw feature maps would be computationally challenging due to the presence of large numbers of feature maps; for instance, a ResNet-50 model produces 2048 distinct 2D feature maps per image at the last layer. For a small validation dataset with only 1K images, this would result in over 2 million 2D feature maps. Strategies for reducing the number of feature maps (e.g. via pruning) can result in the loss of important information. Using ROIAlign and region-level embeddings enables us to reduce dimensionality of feature maps while preserving semantic meaning, enabling efficient automated analysis.
> > > > >
> > > > > (iii) We first clarify that a feature map produced by a model has three dimensions: height $H$, width $W$, and channels/depth $C$. For a ResNet-50 with an input image size of 224x224, the feature map dimensions at the last layer are $H=7$, $W=7$, and $C=2048$. The channels themselves do not explicitly represent image geometry and locations (i.e. the top left or bottom right); rather, each channel corresponds to a 2D feature map of size $7 \times 7$ that characterizes a feature within the input image. These 2D feature maps exhibit significant variations in feature granularity;  the features characterized in each map can range in granularity from global (e.g. color, texture) to local (e.g. individual leaves on a tree). This variability limits the use of raw feature maps for our task. On the other hand, ROIAlign is an effective strategy for embedding specific regions of interest, focusing exclusively on local, fine-grained features.
> > > > >
> > > > > We hope that the above responses address your concerns. Thank you for your time.
> > > > >
> > > > > [1] Tiu, E., et al. Expert-level detection of pathologies from unannotated chest X-ray images via self-supervised learning. Nat. Biomed. Eng (2022).
> > > > >
> > > > > [2] Huang, Z et al. A visual–language foundation model for pathology image analysis using medical Twitter. Nat. Medicine (2023).
> > > > >
> > > > > [3] Zhong et al. RegionCLIP: Region-based Language-Image Pretraining. CVPR 2022.

---

### Official Review · Reviewer_aCX1 · 2024-07-12

**Soundness:** 3
**Presentation:** 3
**Contribution:** 3
**Rating:** 6
**Confidence:** 3

**Summary:**

Vision-language models (VLMs) tend to exhibit poor zero-shot performance when compared to task-specific models. However, fine-tuned VLMs may capture spurious correlations in domain-specific datasets which may be small in size. The paper proposes an automated spurious correlation detection and mitigation method for fine-tuned VLMs. The method first discovers spurious correlations by leveraging a region-level clustering approach to identify precise image features contributing to zero-shot classification errors. Then, it mitigates the identified spurious correlation with a region-aware loss function that enables the VLM to focus on relevant regions and ignore spurious relationships during fine-tuning. Experiments demonstrate the effectiveness of this method on general domain and medical-domain VLMs.

**Strengths:**

- The paper is well-written. Key assumptions are clearly stated and the proposed method is easy to follow.

- The paper proposes a useful method called RAVL to discover and mitigate fine-grained spurious correlations that can be easily interpreted by humans.

- Experiments in the controlled and uncontrolled settings demonstrate that RAVL effectively discovers and mitigates spurious correlations between image features and textual attributes.

**Weaknesses:**

There is a lack of computational complexity analysis. To detect fine-grained spurious correlations, RAVL needs to segment images into multiple regions, cluster all the regions per class, and calculate the contributions of each cluster to mispredictions. Therefore, the computational complexity seems high. It would be better to analyze the time cost of the proposed method.

**Questions:**

1. How to obtain region-level labels when constructing the evaluation dataset (Line 220)?

2. How to determine the hyperparameters used in the spurious correlation discovery and mitigation stages?

**Limitations:**

Yes.

---

> ### Author Rebuttal · Authors · 2024-08-07
>
> We thank Reviewer ACX1 for reviewing our work and providing helpful feedback.
>
> **[Q1] Computational complexity. “There is a lack of computational complexity analysis. To detect fine-grained spurious correlations, RAVL needs to segment images into multiple regions, cluster all the regions per class, and calculate the contributions of each cluster to mispredictions. Therefore, the computational complexity seems high. It would be better to analyze the time cost of the proposed method.”**
>
> We refer the reviewer to General Response [Q1], where we address this point.
>
> **[Q2] Region-level labels for evaluation. “How to obtain region-level labels when constructing the evaluation dataset (Line 220)?”**
>
> In order to evaluate RaVL, we create 654 evaluation settings from two domains: (1) synthetic data (MNIST and FashionMNIST) and (2) real-world data (COCO). Each evaluation setting includes an *evaluation dataset* with images $I_i$, class labels $y_i$, region bounding boxes $R_i$, and region-level labels $L_i$. We obtain region-level labels as follows:
> - For our synthetic data settings, we construct each image $I_i$ to include four equally-sized quadrant regions $\mathcal{R}_i$. We randomly select one quadrant to include the core object (i.e. digit or fashion item) associated with the class label $y_i$. In some images, we randomly select another quadrant to include the pre-defined spurious feature $e^{eval}$, which is a red rectangle in this case. Since these images are synthetically generated, we have a priori knowledge of the features in each quadrant, which we use to generate the region-level label set $L_i$.
> - For our real-world settings, the set $\mathcal{R}_i$ consists of the ground-truth bounding boxes annotated in COCO and $L_i$ consists of the associated object labels. We note that recent advances in open-set object detection (e.g. Recognize Anything) can enable this procedure to be extended to other datasets even if ground-truth bounding boxes and labels are not available.
>
> We additionally emphasize here that the RaVL discovery and mitigation approaches do not require region-level labels, and users who wish to use RaVL do not need access to region-level labels. Here, we exclusively use region-level labels within our evaluation framework in order to quantitatively assess performance.
>
> **[Q3] Hyperparameter selection. “How to determine the hyperparameters used in the spurious correlation discovery and mitigation stages?”**
>
> Hyperparameters for the discovery stage (RaVL Stage 1) are selected as follows:
> - *Number of clusters*: RaVL includes a clustering step to identify groups of visually-similar regions. We do not manually set this hyperparameter; rather, we leverage an automated approach for selecting the optimal number of clusters. For each evaluation setting, we sweep all cluster sizes ranging between $|\mathcal{Y}| * 2$ and $|\mathcal{Y}| * 5$ where $|\mathcal{Y}|$ represents the size of the class label set; we then select the optimal number of clusters using Silhouette scores. We select these bounds to be larger than the class label set size by several multiples in order to ensure that clusters adequately separate distinct features. Prior works have also utilized overclustering approaches for this objective [5,6]. Users can adjust the bounds based on their composition of their dataset; for instance, complex datasets with diverse features may require a larger range.
> - *Threshold for pruning cluster influence scores ($\tau_l$)*: In order to identify candidate image features that directly contribute to classification errors, we prune all clusters with low cluster influence scores below a threshold $\tau_l$. We set $\tau_l$ to 0.25 in this work. This hyperparameter was determined empirically based on experiments on a small set of validation data.
>
> Hyperparameters for the mitigation stage (RaVL Stage 2) are selected as follows:
> - *Contrastive loss temperature $\tau$*: We set the contrastive loss temperature to a default value of 0.07 based on prior work [7].
> - *Loss weighting factor $\lambda$*: The loss weighting factor $\lambda$ balances the tradeoff between the original CLIP loss function $L_{CL}$ and our novel region-aware loss function $L_{R} + L_{A}$. We set $\lambda$ to 0.8 in this work, which upweights the CLIP loss function $L_{CL}$. This weighting factor ensures that the model $\mathcal{M}_{new}$ learns accurate global image-text relationships without capturing fine-grained spurious correlations. We determined this hyperparameter based on experiments on a small set of validation data. We additionally note here that the "Upsampled FT" baseline in Table 3 is an ablation where $\lambda$ is set to 1 (and the region-aware loss function is not utilized); this demonstrates the efficacy of our loss function.
>
> We note that like all hyperparameters, these values could likely be further optimized; however, even without extensive hyperparameter tuning, we achieve competitive results. Additional details on hyperparameters are provided in Appendix B, C, and D.
>
> We again thank Reviewer ACX1 for their review of our manuscript and their positive overall assessment of our work. We hope that the above responses adequately address all concerns.

---

> > ### Comment · Reviewer_aCX1 · 2024-08-10
> >
> > Thanks for your detailed responses. My concerns are well addressed.

---

### Official Review · Reviewer_LC5i · 2024-07-12

**Soundness:** 3
**Presentation:** 3
**Contribution:** 3
**Rating:** 6
**Confidence:** 4

**Summary:**

This paper tackles spurious correlations between image features and textual attributes in fine-tuned VLMs. It proposes an approach to discover and mitigate spurious correlations using local image features (image regions rather than a whole image). Experiments are done in both controlled settings and realistic settings.

**Strengths:**

Discovering and mitigating spurious correlations is an interesting problem. The approach of looking into image regions is sound.
The paper is generally well written.

**Weaknesses:**

1) Looking into the regional features may make the approach computationally expensive. Computational complexity analysis is missing in the paper.
2) A lot of experiments were done under controlled settings. More evaluations in the wild would make the results stronger.
3) A small error in writing, line 259 says “are designed for unimodal settings” - ref [50] is designed for multi-modal settings.

**Questions:**

Have you tried the approach on other settings, other than fine-tuned VLM?  For example, spurious correlations can happen in other vision classification models as well.

**Limitations:**

There are discussions on limitations in the supplementary material.

---

> ### Author Rebuttal · Authors · 2024-08-07
>
> We thank Reviewer LC5I for reviewing our work and providing helpful feedback.
>
> **[Q1] Computational complexity. “Looking into the regional features may make the approach computationally expensive. Computational complexity analysis is missing in the paper.”**
>
> We refer the reviewer to General Response [Q1], where we address this point.
>
> **[Q2] In-the-wild experiments. “A lot of experiments were done under controlled settings. More evaluations in the wild would make the results stronger.”**
>
> Thank you for this suggestion, and we agree about the importance of real-world validation. However, we note that it is challenging to quantitatively evaluate the accuracy of discovery and mitigation approaches, since the ground-truth spurious correlations learned by a VLM are typically unknown. As a result, we opted to use controlled evaluations, which artificially induce spurious correlations and then quantitatively evaluate the extent to which the correlation can be detected and mitigated. Importantly, using controlled evaluations enables us to **perform large-scale, quantitative evaluations of RaVL without using humans in the loop**. We believe that this procedure is essential before scaling to unlabeled real-world datasets.
>
> In addition to these controlled quantitative evaluations, our original manuscript provided qualitative results from several in-the-wild evaluations (Section 3.3 and Figure 3). In response to the reviewer's suggestion for expanding these analyses, we have extended this analysis with additional qualitative evaluations. We refer the reviewer to the PDF provided with the general response above. In Rebuttal Figure 1, we show the following:
> - For the OpenCLIP ViT-L/14 model, RaVL surfaces a feature cluster consisting of green plants and fences. We observe a performance gap of 18.3 points between images with class label *"outdoor chicken coop"* that contain the RaVL-identified feature and those that do not contain the feature. This suggests that the OpenCLIP ViT-L/14 model can better classify an *"outdoor chicken coop"* scene when green plants and fences are present.
> - For the CLIP ViT-B/32 model, RaVL surfaces a feature cluster consisting of people. We observe a performance gap of 24.3 points between images with class label *"pub (indoor)"* that contain the RaVL-identified feature and those that do not contain the feature. This suggests that the CLIP ViT-B/32 model can better classify a *"pub (indoor)"* scene when people are present.
> - For the OpenCLIP ResNet-101 model, RaVL surfaces a feature cluster consisting of chairs. We observe a performance gap of 23.3 points between images with class label *"restaurant patio"* that contain the RaVL-identified feature and those that do not contain the feature. This suggests that the OpenCLIP ResNet-101 model can better classify *"restaurant patio"* scenes when chairs are present.
>
> **[Q3] References. “A small error in writing, line 259 says “are designed for unimodal settings” - ref [50] is designed for multi-modal settings.”**
>
> Thank you for raising this point. In Line 259, the phrase "designed for unimodal settings" is intended to refer solely to Distilling Failures, George, and Domino (referenced in Line 257). We agree with the reviewer that Spurious-Aware Detection (reference [50]) is not unimodal, and we will be sure to improve the clarity of this sentence in the final version of our manuscript.
>
> **[Q4] Extending beyond VLMs. “Have you tried the approach on other settings, other than fine-tuned VLM? For example, spurious correlations can happen in other vision classification models as well.”**
>
> Thank you for this suggestion, and we agree that spurious correlations can occur in many types of models. In this work, we focus specifically on the setting of fine-tuned vision-language models because fine-tuned VLMs (i) are becoming increasingly commonplace, particularly in domain-specific applications like medicine, (ii) are likely to be trained on small domain-specific datasets, preventing models from gaining the robustness benefits associated with web-scale training and increasing the likelihood of learning spurious correlations, and (iii) are an underresearched yet important problem setting. In order to address spurious correlations in the fine-tuned VLM setting, RaVL assumes the existence of paired language; consequently, RaVL utilizes text embeddings for both discovery (RaVL Stage 1) and mitigation (RaVL Stage 2).
>
> In the context of vision-only classification models where the language modality is not present, our discovery approach can be modified by computing the image score distribution vector $s_{I_i}$ and the region score distribution matrix $S_{R_i}$ using softmax-normalized class logits. The mitigation approach can be modified by framing the region-aware contrastive loss function as a cross entropy loss between region-level logits and class labels. We aim to explore these directions in future work.
>
> We again thank Reviewer LC5I for their review of our manuscript and their positive overall assessment of our work. We hope that the above responses adequately address all concerns.

---

> > ### Comment · Reviewer_LC5i · 2024-08-12
> >
> > Thanks for the detailed rebuttal.

---

> ### Comment · Reviewer_LC5i · 2024-08-13
>
> Thanks again for the rebuttal.
>
> This paper is based on the assumption that, “a model M that has learned a spurious correlation between an image feature e_a and a textual attribute y will demonstrate low zero-shot performance on (i) images in D_V with label y without the feature e_a and (ii) images in D_V with other labels Y \ {y} with the feature e_a. ”  (e.g., see lines 124 - 127 in the paper)
>
> I have one more question on the paper.  For some “true” image features, when the following are true, they may also appear to have the property (e.g., see lines 124 - 127 in the paper) mentioned above.
> 1)  The “true” image features only appear in a subset of the class y. This can happen because not all images in the same class have the same features.
> 2) The training data and test data are sampled differently: the “true” features only appear (or appear more often) in the training data with class y, and appear less often in the test data with class y.
>
> How do you distinguish between spurious image features and the “true” image features with different appearing frequencies in training and test data?

---

> > ### Author Response · Authors · 2024-08-14
> > **Response to Reviewer LC5i**
> >
> > Below, we respond to the points raised by Reviewer LC5i during the discussion period.
> >
> > **Definition of spurious correlations. “For some “true” image features, when the following are true, they may also appear to have the property (e.g., see lines 124 - 127 in the paper) mentioned above. (1) The “true” image features only appear in a subset of the class y. This can happen because not all images in the same class have the same features. (2) The training data and test data are sampled differently: the “true” features only appear (or appear more often) in the training data with class y, and appear less often in the test data with class y. How do you distinguish between spurious image features and the “true” image features with different appearing frequencies in training and test data?”**
> >
> > Our definition of spurious correlations is consistent with prior work (e.g. [1,2]). In reference to the specific cases listed by the reviewer above, it is standard practice to consider a "true" image feature as one that appears consistently within the class. This means that “true” features will (1) appear in all images rather than just a subset and (2) will be consistently associated with the class label $y$ at both training and test time. For example, when considering images from the class “bird”, it is reasonable to assume that all images will include the true features of feathers, wings, and a beak, which are defining characteristics of birds. This will be true for all birds, independent of whether they are in a training or testing dataset.
> >
> > We refer the reviewer to Singla et al. [2], which formally defines true features as those that are "always a part of the" class definition; meanwhile, spurious features are defined as those that "are likely to co-occur" with the true features but are not a part of the class definition. We follow these definitions in our work. We demonstrate with 654 evaluations in both synthetic and real-world settings that our approach can differentiate between spurious features and true features, as shown in Figure 2 and Table 1; we corroborate this further with in-the-wild experiments.
> >
> > We hope that this clarifies the definitions of true and spurious features.
> >
> > [1] Eyuboglu et al. “Domino: Discovering Systematic Errors with Cross-Modal Embeddings” ICLR 2022.
> >
> > [2] Singla, S. et al. Salient ImageNet: How to discover spurious features in Deep Learning? ICLR, 2022.

---

> ### Comment · Reviewer_LC5i · 2024-08-14
>
> Thanks for the response. My concerns have been well addressed.

---

### Author Rebuttal · Authors · 2024-08-07

We thank the reviewers for their thoughtful review of our manuscript. We were encouraged to see that all reviewers rated our work as a "technically solid, moderate-to-high impact paper". Reviewers also found the paper to be "well-written" and "well-structured" (Reviewers LC5I, ACX1, ONXK); the problem setting to be "interesting" (Reviewer LC5I); and our proposed approach to be "sound", "novel", and "easy to follow" with "notable" performance improvements (Reviewers LC5I, ONXK, ACX1).

In response to feedback, we provide a general response here to points raised by multiple reviewers, individual responses below to address each reviewer’s concerns, and an attached one-page PDF with new figures.

**[Q1] Reviewers LC5I and ACX1 asked for additional analysis on the computational complexity of RaVL.**

We designed our approach to be computationally inexpensive; in particular, the discovery stage can be run efficiently on CPU and the mitigation stage adds only a small computational overhead. Although RaVL does involve multiple stages as noted by reviewer ACX1, our procedure delivers the significant advantage of enabling **fully-automated analysis** of spurious correlations; in contrast, several recent works on fine-grained robustness (both in the vision-only and vision-language settings) have leveraged humans-in-the-loop [1,2,3]. Below, we provide an analysis of computational complexity for each stage of RaVL.

*Computational complexity analysis of RaVL Stage 1*: In response to points from Reviewer ACX1 about the discovery stage (RaVL Stage 1), we note that our approach is specifically designed to be run on a labeled validation dataset $\mathcal{D_V}$; in real-world settings, validation datasets are often relatively small in size due to the human effort needed for securing labels, rendering this stage as computationally inexpensive for diverse applications. Even if the validation dataset is large in size, RaVL operates efficiently as follows:
- First, RaVL preprocesses images by decomposing each image into candidate regions; there are a variety of ways in which a user can decompose an image into regions, such as by using equal-sized segments (e.g. quadrants) or running inference with region proposal networks (RPNs). Both methods are inexpensive and only need to be run once in an offline manner. Similar approaches have been applied to large-scale datasets in prior work [4].
- Then, embeddings need to be generated for each region, which can be done by utilizing VLM $\mathcal{M}$ for inference (forward passes only). Across a set of 10 FashionMNIST and COCO evaluation settings, we observe embedding generation to take a mean of 24.5 seconds on a single A100 GPU.
- Finally, given candidate regions and corresponding embeddings, the remainder of the RaVL discovery procedure (clustering and computation of metrics) can be run completely on CPU. Across a set of 10 evaluation settings on COCO and FashionMNIST, we observe that clustering and computation of metrics require a mean of 3.4 seconds to run on a single A100 GPU.

*Computational complexity analysis of RaVL Stage 2*: The mitigation stage (RaVL Stage 2) requires finetuning a VLM $M_{new}$. Across a set of 10 evaluation settings on COCO and FashionMNIST, we observe that the inclusion of our fine-grained region-aware loss function at this stage adds an average of 0.15 seconds per training step (on a single A100 GPU) in comparison to the original fine-tuning procedure for $M$.

We will be sure to update our manuscript with this computational complexity analysis.

We would again like to thank all reviewers for their time and feedback, and we hope that our responses adequately address all concerns.

**References:**

[1] Yang, Y., et al. Mitigating Spurious Correlations in Multi-modal Models during Fine-tuning. ICML, 2023.

[2] Singla, S. et al. Salient ImageNet: How to discover spurious features in Deep Learning? ICLR, 2022.

[3] Moayeri, M. et al. Hard ImageNet: Segmentations for Objects with Strong Spurious Cues. NeurIps, 2022.

[4] Zhong, Y. et al. RegionCLIP: Region-based Language-Image Pretraining. CVPR, 2022.

[5] Eyuboglu et al. “Domino: Discovering Systematic Errors with Cross-Modal Embeddings.” ICLR 2022.

[6] Sohoni et al. “No subclass left behind: Fine-grained robustness in coarse-grained classification problems.” NeurIps 2020.

[7] Radford et al. “Learning Transferable Visual Models From Natural Language Supervision.” ICML 2021.

[8] Winkler, J. K., et al. Association between surgical skin markings in dermoscopic images and diagnostic performance of a deep learning convolutional neural network for melanoma recognition. JAMA dermatology, 2019.

[9] Oakden-Rayner, L., et al. Hidden stratification causes clinically meaningful failures
in machine learning for medical imaging. ACM Conference on Health, Inference, and Learning (CHIL), 2020.

[10] DeGrave et al. AI for radiographic COVID-19 detection selects shortcuts over signal. Nature Machine Intelligence, 2021.

[11] Moayeri et al. Spuriosity Rankings: Sorting Data to Measure and Mitigate Biases. NeurIps, 2023.

[12] Chen et al. “Why do We Need Large Batchsizes in Contrastive Learning? A Gradient-Bias Perspective.” NeurIps 2022.

---

### Decision · Program_Chairs · 2024-09-25

**Decision:**

Accept (poster)

**Comment:**

RaVL is a method to detect and mitigate spurious correlations in VLMs. The method tries to figure out what features contribute to zero-shot classification errors and then try to remedy those. The reviewers agree this is an important problem, the paper is well written and presents a contribution to the literature. There was some discussion with one of the reviewers about the impact of batch size on performance, but I do not think this changes the main findings of the paper. Overall the method seems solid, experimental evidence seems satisfactory. For these reasons I vote to accept this paper.